# Earthquake transformer—an attentive deep-learning model for simultaneous earthquake detection and phase picking

S. Mostafa Mousavi 1✉, William L. Ellsworth 1, Weiqiang Zhu 1, Lindsay Y. Chuang2 & Gregory C. Beroza1

Earthquake signal detection and seismic phase picking are challenging tasks in the processing of noisy data and the monitoring of microearthquakes. Here we present a global deep-learning model for simultaneous earthquake detection and phase picking. Performing these two related tasks in tandem improves model performance in each individual task by combining information in phases and in the full waveform of earthquake signals by using a hierarchical attention mechanism. We show that our model outperforms previous deep-learning and traditional phase-picking and detection algorithms. Applying our model to 5 weeks of continuous data recorded during 2000 Tottori earthquakes in Japan, we were able to detect and locate two times more earthquakes using only a portion (less than 1/3) of seismic stations. Our model picks P and S phases with precision close to manual picks by human analysts; however, its high efficiency and higher sensitivity can result in detecting and characterizing more and smaller events.

[1] Geophysics Department, Stanford University, 397 Panama Mall, Stanford, CA 94305-2215, USA. [2] School of Earth and Atmospheric Sciences, Georgia Institute of Technology, Atlanta, GA 30332, USA. ✉email: mmousavi@stanford.edu

Deep learning is a widely applied and effective method for a broad range of applications[1]. Earthquake monitoring has a growing need for more efficient and robust tools for processing of increasingly large data volumes, is conceptually straightforward, and has a large quantity of available labeled data, which make earthquake detection and phase picking attractive targets for the new wave of machine learning applications in seismology. To date, earthquake signal detection and phase-picking form the largest portion of this relatively young sub-field[2–10]. Despite the differences in approaches and results, most of these studies find important advantages to deep-learning-based methods compared with traditional approaches[11,12].

Earthquake signal detection and phase picking are challenging problems in earthquake monitoring. Detection refers to identification of earthquake signals among a wide variety of non-earthquake signals and noise recorded by a seismic sensor. Phase picking is the measurement of arrival times of distinct seismic phases (P-wave and S-wave phases) within an earthquake signal that are used to estimate the location of an earthquake. Although these two tasks share some similarities, their objectives are not quite the same. Minimizing the false negative and false positive rates are the main goals in detection; however, in phase picking the focus is on increasing the temporal accuracy of arrival-time picks. This is due to the extreme sensitivity of earthquake location estimates to earthquake arrival time measurements - 0.01 second of error in determining P-wave arrivals can translate to tens of meters of error in location. Although both detection and picking can be viewed as identifying distinct variations in time-series data, phase picking is a local problem compared to detection, which uses a more global view of the full waveform and consists of information from multiple seismic phases including scattered waves. Because of this, previous machine-learning studies have approached these tasks individually using separate networks; however, these tasks are closely related to each other. In practice, analysts first look at the entire waveform on multiple stations to identify consistent elements of an earthquake signal (e.g. P, S, coda and surface waves) with a specific ordering (P-wave always arrives before S-wave, higher frequency body waves always precede dispersive surface waves etc.) to determine whether or not a signal is from an earthquake. Then they focus on each phase to pick the arrival times precisely. This practice indicates the interconnection of these two tasks and the importance of contextual information in earthquake signal modeling.

Deep-learning detection/picking models work by learning general characteristics of earthquake waveforms and seismic phases from high-level representations. Here we test the hypothesis that better representations obtained by incorporating the contextual information in earthquake waveforms will result in better models. Our expectation is that not all parts of a seismic signal are equally relevant for a specific classification task. Hence, it is beneficial to determine the relevant sections for modeling the interaction of local (narrow windows around specific phase arrivals) and global (full waveform) seismic features. We achieve this by incorporating an attention mechanism[13] into our network. Attention mechanisms in Neural Networks are inspired by human visual attention. Humans focus on a certain region of an image with high resolution while perceiving the surrounding image at low resolution and then adjusting the focal point over time. Our model emulates this through two levels of attention mechanism, one at the global level for identifying an earthquake signal in the input time series, and one at the local level for identifying different seismic phases within that earthquake signal.

We introduce a new deep-learing model (EQTransformer[1]) for the simultaneous detection of earthquake signals and picking first P and S phases on single-station data recorded at local epicentral distances (<300 km) based on attention mechanism. Here we present our approach and compare its performance with multiple deep-learning and traditional pickers and detectors. Our trained model is applied on 5 weeks of continuous waveforms recorded in Japan. The events we detect are located to demonstrate the generalization of the model to other regions and its ability to improve earthquake source characterization.

## Results

**Network architecture**. Our neural network has a multi-task structure consisting of one very-deep encoder and three separate decoders composed of 1D convolutions, bi-directional and uni-directional long-short-term memories (LSTM), Network-in-Network, residual connections, feed-forward layers, transformer, and self-attentive layers (Fig. 1). More details are provided in the method section. The encoder consumes the seismic signals in the time domain and generates a high-level representation and contextual information on their temporal dependencies. Decoders then use this information to map the high-level features to three sequences of probabilities associated with: existence of an earthquake signal, P-phase, and S-phase, for each time point.

In self-attentive models the amount of memory grows with respect to the sequence length; hence, we add a down-sampling section composed of convolutional and max-pooling layers to the front of the encoder. These down-sampled features are transformed to high-level representations through a series of residual convolution and LSTM blocks. A global attention section at the end of the encoder aims at directing the attention of the network to the parts associated with the earthquake signal. These high-level features are then directly mapped to a vector of probabilities representing the existence of an earthquake signal (detection) using one decoder branch. Two other decoder branches are associated with the P-phase and the S-phase respectively in which an LSTM/local attention unit is placed at the beginning. This local attention will further direct the attention of the network into local features within the earthquake waveform that are associated with individual seismic phases. Residual connections within each block and techniques such as network-in-networks help to expand the depth of the network while keeping the error rate and training speed manageable. As a result, our very deep network with 56 layers has only about 372 K trainable parameters. The network architecture design is based on domain expertise. Optimization and hyperparameter selection are based on experiments on a large number of prototype networks.

**Data and labeling**. We used STanford EArthquake Dataset (STEAD)[13] to train the network. STEAD is a large-scale global dataset of labeled earthquake and non-earthquake signals. Here we used 1 M earthquake and 300 K noise waveforms (including both ambient and cultural noise) recorded by seismic stations at epicentral distances up to 300 km. Earthquake waveforms are associated with about 450 K earthquakes with a diverse geographical distribution (Fig. 2). The majority of these earthquakes are smaller than M 2.5 and have been recorded within 100 km from the epicenter. A full description of properties of the dataset can be found in[13]. Although STEAD contains earthquake waveforms from a variety of geographical regions and tectonic settings, it does not have any earthquake seismograms from Japan. We split the data into training (85%), validation (5%), and test (10%) sets randomly. Waveforms are 1 minute long with a sampling rate of 100 Hz and are causally band-passed filtered from 1.0–45.0 Hz. A box-shaped label is used as ground truth for the detection. In this binary vector, corresponding samples from the P arrival to the S arrival + 1.4 × (S - P time) are set to 1 and the rest to 0. We tested three different forms of: box, Gaussian, and triangular, to label phase arrivals. Triangular labeling resulted in a lower loss

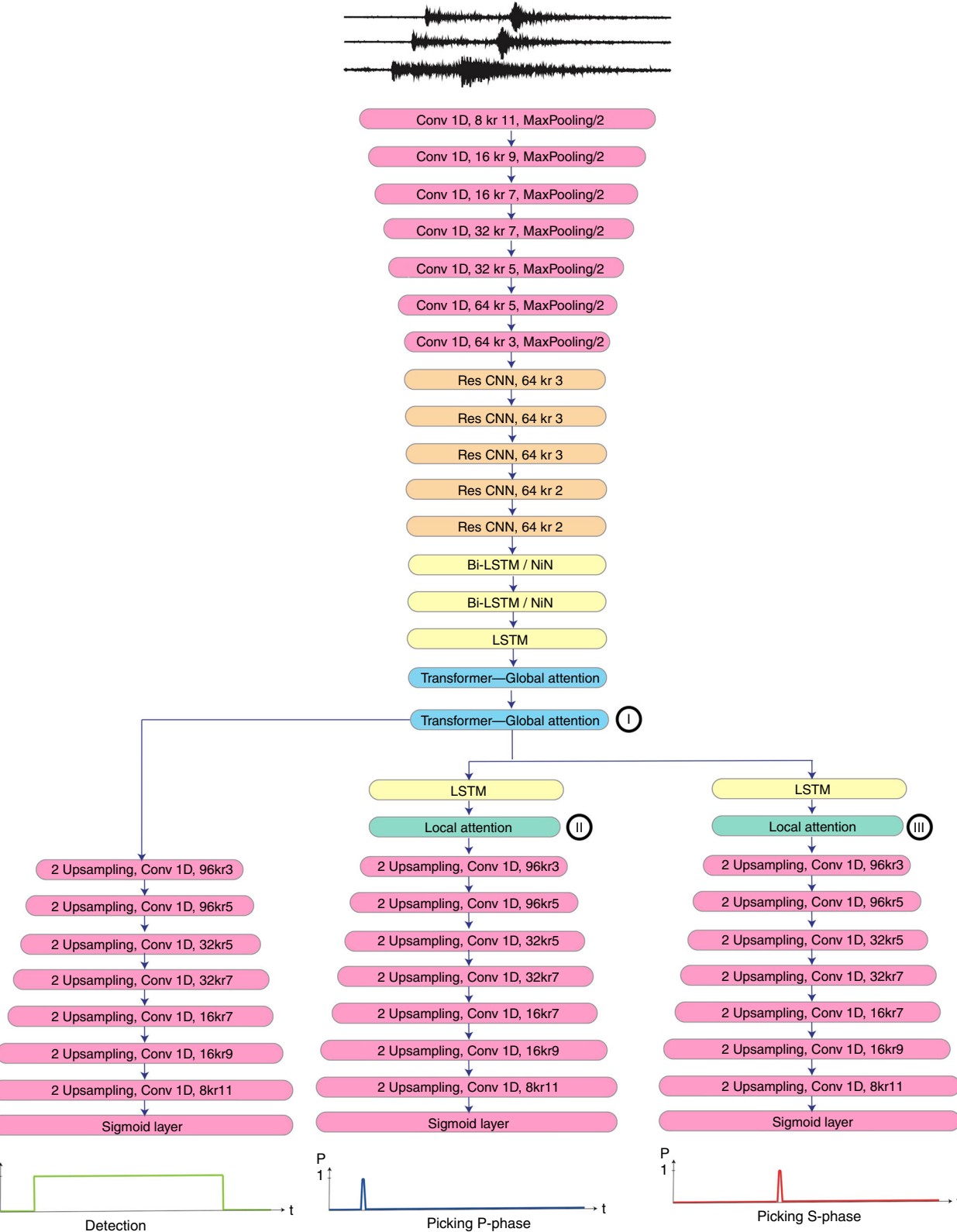

**Fig. 1 Network architecture.** Our network architecture. Full details of each block are provided in the method section. The convolutional layers read as (number of kernels) kr (kernel size).

and higher F-score during our hyperparameter selection procedure and is used for the final model. In this form, probabilities of P and S are set to 1 at the first arriving P and S wave and linearly decrease to 0 within 20 samples before and 20 samples after each phase arrival.

**Training**. For both convolutional and LSTM units, all the weight and filter matrices were initialized with a Xavier normal initializer[14] and bias vectors set to zeros. We used ADAM[15] with varying learning rates for optimization while the learning rate varied during training. The model took O(89) hours to complete

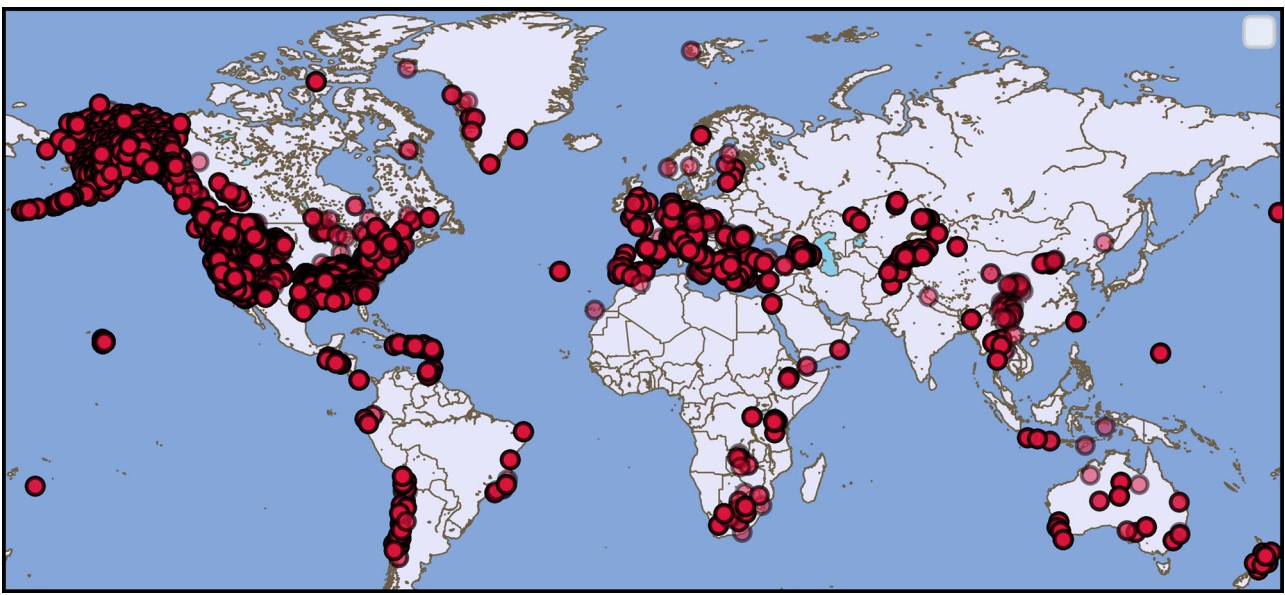

**Fig. 2 The training and test dataset.** Geographic distribution of station locations recording 300 k noise and 1 M earthquake seismograms in STanford EArthquake Dataset (STEAD) used in this study.

the training using 4 parallel Tesla-V100 GPUs under the tensorflow framework[16]. Training stopped when validation loss did not improve for 12 consecutive epochs.The data was augmented by adding a secondary earthquake signal into the empty part of the trace, adding a random level of Gaussian noise into earthquake waveform, randomly shifting the event within the trace - through array rotation, randomly adding gaps (zeroing a short time span) in noise waveforms, and randomly dropping one or two channels (zeroing values of one or two channels) with 0.3, 0.5, 0.99, 0.2 and 0.3 probability respectively. Half of the data in each batch are augmented versions of the waveforms in the other half. Data augmentation and normalization (by standard deviation) are done simultaneously during the training on 24 CPUs in parallel. We used a dropout rate of 0.1 for all dropout layers both at training and test time.

**Exploring the network's attention**. The attention weights define how much of each input state should be considered for predicting each output and can be interpreted as a vector of importance weights. By explicitly visualizing these attention weights we can see on what parts of the input sequence the neural network has learned to focus.

Figure 3 presents the output of each of these attention layers (summation of hidden states at all other time steps, weighted by their scoring) for one specific event from the evaluation set. We can see that the network has learned to focus on different parts of the waveform at different attention levels. This highlights the most useful parts of the input waveform for each task. The shorter path through the detection decoder and its higher loss (due to longer length of label) naturally force the network to first learn to distinguish the earthquake signal within a time series. We can see this from the learning curves as well (Supplementary Fig. 2). This mimics a seismic analyst's decision-making workflow. The second transformer (I in Fig. 1), at the end of encoder section, mainly passes the information corresponding to the earthquake signal to the subsequent decoders. This means that the encoder learns to select what parts of the signal holds the most important information for detection and phase picking. This information is directly used by the detection decoder to predict the existence of an earthquake signal in the time series. The local attention layers at the beginning of P and S decoders further focus on

smaller sections, within the earthquake waveform, to make their predictions. The alignment scores are normalized and can be thought of as probability distributions. So we can interpret the hierarchical attention mechanisms in our network as conditional probabilities: $P(earthquake signal | input waveform) = encoder(input waveform)$, and $P(P\_phase | input waveform) = P\_decoder(P(earthquake signal | input waveform))$.

**Results and comparison with other methods**. We used more than 113 k test waveforms (both earthquake and noise examples) to evaluate and to compare the detection and picking performance of EQTransformer with other deep-learning and traditional methods. Deep-learning models used here for the comparisons are pre-trained models based on different training sets and all are applied to a common test set from STEAD. The test set data contains 1-min long 3C-waveforms. All the tests are performed without additional filtering of the test data. Figure 4 illustrates the network predictions for 4 representative samples from the test set (Fig. 4a–d). The model works very well for earthquakes with different waveform shapes. The model is able to retain a global view for the detection while picking distinct arrival times with high temporal resolution. This can be seen clearly from the example in Fig. 4b, where two strong and apparently separate waves are detected as parts of a single event rather than two individual events. The very deep structure of the network makes it less sensitive to the noise level and it works well for small events with a high background noise (Fig. 4c, d). Moreover, the provided uncertainties can be useful to identify unreliable predictions even when the output probabilities are high (Fig. 4c).

We also applied the model to continuous data. The only reprocessing steps that need to be done prior to the test/prediction are: filling the gaps, removing the trend, band-pass filtering, and re-sampling the data to 100 Hz. Augmentations are applied only during the training process. After pre-processing, the continuous data can be sliced into 1-min windows (preferentially with some overlap). The model can be applied on a single or a batch of these 1-min slices. The normalization is done during feeding the data to the model. Figure 4e–h presents the results of application of the model to continuous data recorded in Ridgecrest, California and Tottori, Japan.

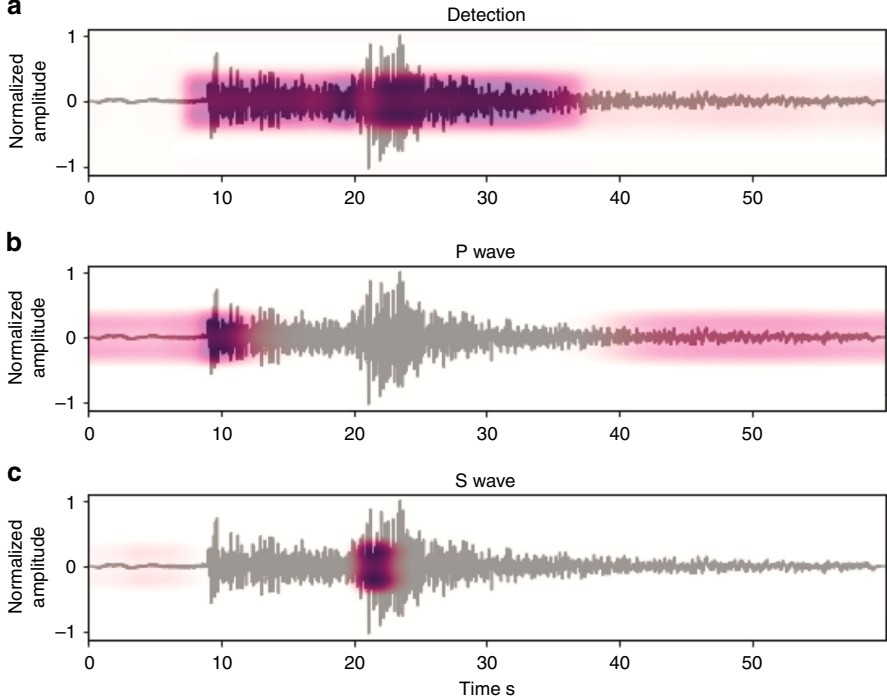

**Fig. 3 Attention weights.** Input waveform overlain by contextual information - output of the attention layers for **a** transformer (I in Fig 1), **b** local attention for P-phase (II in Fig 1), and **c** the local attention for S-phase (III in Fig 1).

The augmentations used during the training process were essential in the performance of the final model. P arrival times are clustered at the first quarter of the windows in the training data and each trace only contains 1 event. However, Fig. 4e, f shows that regardless of these characteristics in the training set, the model works well when more than one event exists in a 1-min window and at various time points. The model can detect/pick events that occur on the edges (Fig. 4e) as long as 0.2 s of P and S exist within the window. Augmentations prevent the model from producing false positives at abrupt changes due filling the gaps in continuous data (Fig. 3e). Our model works for single channel data (Fig. 3h) or when other channels are broken or dominated by noise (Fig. 3g). More examples of the model performance on a variety of cases are provided in the supplementary materials (Supplementary Fig. 3–6).

We present the detection performance on the test set with a confusion matrix (Supplementary Fig. 7). Our method resulted in only 1 false positive with 0 false negatives (no missing events) out of 113 K test samples using a threshold value of 0.5 (Supplementary Fig. 8). To compare the performances, we applied three deep-learning (DetNet[5], Yews[4], and CRED[7]) detectors and one traditional (STA/LTA[11]) detector to the same test set (Table 1). We should acknowledge that there is a level of tuning involved in each of these approaches (traditional and deep-learning detectors/pickers), and that the performance can vary based on this tuning. Our proposed model outperforms the other methods in terms of F1-score. CRED also contains both convolutional and recurrent units and was trained on the same data set (STEAD); however, its performance did not reach that of EQTransformer. This points to the beneficial effects of the incorporated attention mechanism and the use of a deeper network for earthquake signal detection. DetNet was trained on a much smaller dataset compare to Yews, but it has a better performance; however, neither DetNet nor Yews reach the STA/LTA results in terms of F-score, and STA/LTA requires no training.

We now compare the picking performance with five deep-learning (PhaseNet[8], GPD[10], PpkNet[5], Yews[4], PickNet[2])

(Supplementary Fig. 9) and three traditional (Kurtosis[17], Filter-Picker[18], and AIC[19]) (Supplementary Fig. 10) auto pickers. We did not find a well-documented code or trained model for other deep-learning pickers mentioned in section 2. These are pre-trained models based on datasets of different sizes and from different regions to evaluate their generalization. The list of these training sets are given in Tables 2 and 3 for P and S picks. We assess the performance of each picker using 7 scores (standard deviation of error, mean error, precision, recall, F1-score, mean absolute error, and mean absolute percentage error). A pick was considered as a true positive when its absolute distance from the ground truth was less than 0.5 second. EQTransformer increases the F-scores of both P and S picking. The improvement in P-wave picks are more significant than for S-wave picks. This may be due the fact that picking S-waves is more difficult and prone to more errors, which can lead to higher labeling error in the training set. The error distributions for some of the deep-learning pickers are not uniform and cluster at sporadic times perhaps due to their moving-window scheme. All of these models (GPD[10], PpkNet[5], and Yews[4]) use wider labels compared to the other models (PhaseNet[8], PickNet[2], and EQTransformer). However, it is difficult to narrow down the exact reason behind their non-normal error distributions.

**Application to other regions**. STEAD, the dataset used for training of our model, does not contain any waveform data from Japan. This makes Japan an ideal place to test the performance and generalization of our model. We select the aftershock region of the 2000 $M_w$ 6.6 western Tottori earthquake for this test. Our detector/phase-picker model was applied to continuous data of 18 HiNet stations from 6 October to 17 November 2000. These are a portion of stations (57) originally used for studying this sequence by the Japan Meteorological Agency (JMA). The prediction module in EQTransformer code outputs the results when at least one phase (P or S) with a probability above a specified threshold values exists over a time period with high probability of being an

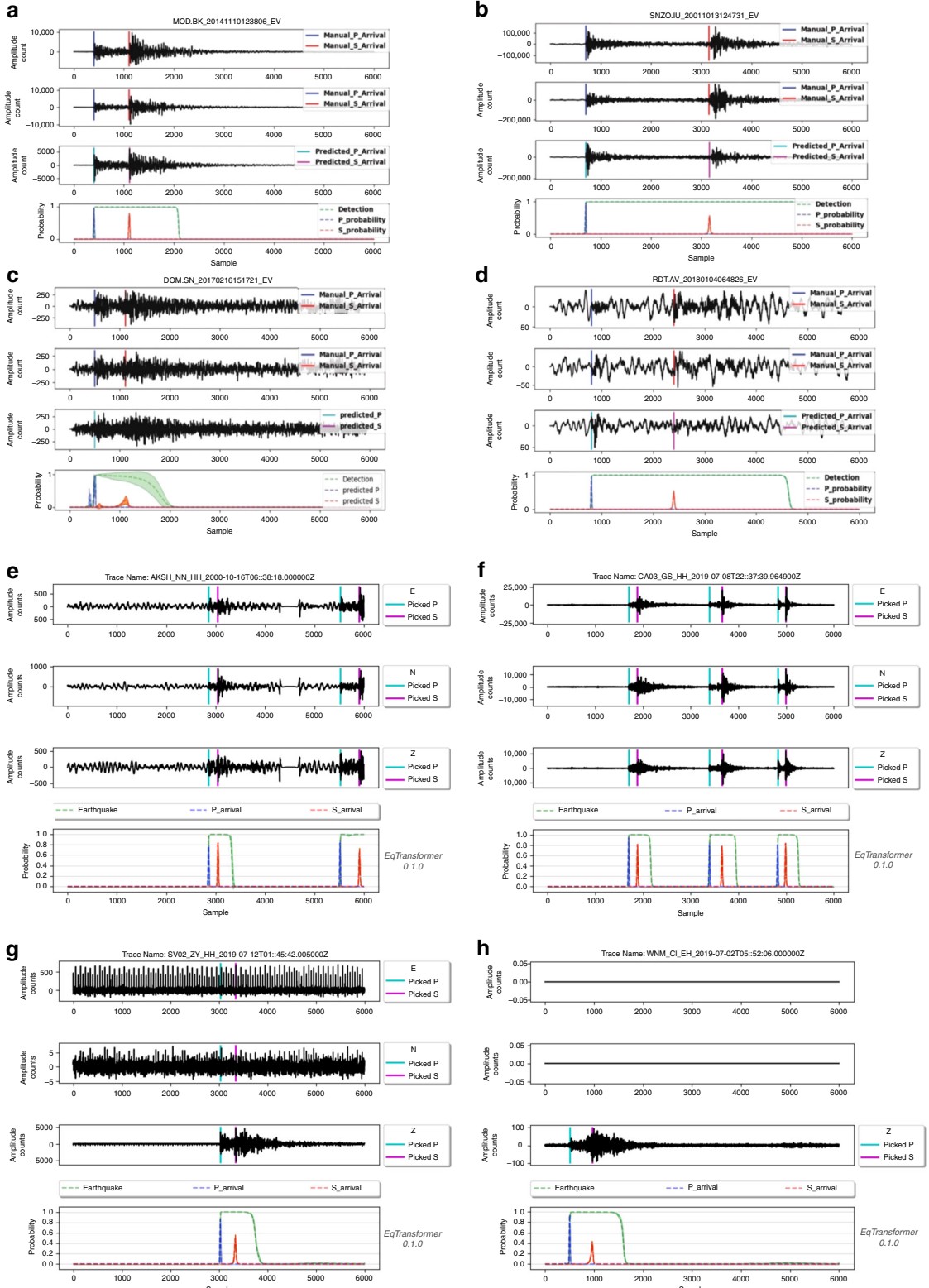

**Fig. 4 Test results.** Four representative waveforms (**a**–**d**) from the test set and four waveforms from applying the model to continuous data in Tottori, Japan (**e**) and Redgcrest, California (**f**–**h**) presenting performance of the model on different types of data. Each waveform is 60 seconds long with 100 samples per second band-bass filtered from 1-45 Hz. Each panel shows three channel waveforms on top and output predictions of the model for earthquake signal detection, P-arrival, and S-arrival at the bottom. In **a** to **d**, the vertical color-coded lines on first two channels are manual arrival time picks from catalogs. **a** is an event with the local magnitude of 2.2 recorded in 55 km distance from the epicenter, **b** is an event with the local magnitude of 4.3 recorded 173 km away from the epicenter, **c** is an event with the local magnitude of 0.1 recorded 38 km away from the epicenter, and **d** is an event with the local magnitude of 2.0 recorded 110 km away from the epicenter. The output probabilities are presented as distributions that can represent variations or model uncertainties. **e** to **h** are detected events after applying the model to continuous data, representing the performance of the model when more than one event exists in a 1-minute window (**e** and **f**), when data contains gaps filled by zeros (**e**), when an event occurs near the edge (**e**), when two channels are broken or noisy (**g**), or when only one component data exists (**h**).

**Table 1 Detection performance.**

| Model | Pr | Re | F1 | Training data | Training size | Ref. |
|---|---|---|---|---|---|---|
| EQTransformer | **1.0** | **1.0** | **1.0** | Global | 1.2M | This study |
| CRED | **1.0** | 0.96 | 0.98 | Global | 1.2M | 7 |
| DetNet | **1.0** | 0.89 | 0.94 | China | 30K | 5 |
| Yews | 0.84 | 0.85 | 0.85 | Taiwan | 1.4M | 4 |
| STA/LTA | 0.91 | **1.0** | 0.95 | — | — | 11 |

Pr, Re, and F1 are precision, recall, and F1-score respectively. EQTransformer and CRED have been trained on STEAD dataset while DetNet and Yews results are based on pre-trained models on different datasets. Recursive STA/LTA algorithm is used here.
Bold values represent the best performance.

**Table 2 P-phase picking.**

| Model | μ | σ | Pr | Re | F1 | MAE | MAPE | Training data | Training size | Ref. |
|---|---|---|---|---|---|---|---|---|---|---|
| EQTransformer | **0.00** | **0.03** | **0.99** | **0.99** | **0.99** | **0.01** | **0.00** | Global | 1.2M | This study |
| PhaseNet | −0.02 | 0.08 | 0.96 | 0.96 | 0.96 | 0.07 | 0.01 | North California | 780K | 8 |
| GPD | 0.03 | 0.10 | 0.81 | 0.80 | 0.81 | 0.08 | 0.01 | South California | 4.5M | 10 |
| PickNet | **0.00** | 0.09 | 0.81 | 0.49 | 0.61 | 0.07 | 0.02 | Japan | 740K | 2 |
| PpkNet | −0.01 | 0.15 | 0.90 | 0.90 | 0.90 | 0.10 | 1.90 | Japan | 30K | 5 |
| Yews | 0.07 | 0.13 | 0.54 | 0.72 | 0.61 | 0.09 | 0.02 | Taiwan | 1.4M | 4 |
| Kurtosis | −0.03 | 0.09 | 0.94 | 0.79 | 0.86 | 0.08 | 0.01 | — | — | 17 |
| FilterPicker | −0.01 | 0.08 | 0.95 | 0.82 | 0.88 | 0.14 | 0.02 | — | — | 18 |
| AIC | −0.04 | 0.09 | 0.92 | 0.83 | 0.87 | 0.09 | 0.01 | — | — | 19 |

μ and σ are mean and standard deviation of errors (ground truth—prediction) in seconds respectively. Pr, Re, and F1 are precision, recall, and F1-score respectively. MAE and MAPE are mean absolute error and mean absolute percent error respectively. Note Yews and PpkNet models used here are trained based on different datasets mentioned in the related work section.
Bold values represent the best performance.

**Table 3 S-phase picking.**

| Model | μ | σ | Pr | Re | F1 | MAE | MAPE | Training data | Training size | Ref. |
|---|---|---|---|---|---|---|---|---|---|---|
| EQTransformer | **0.00** | **0.11** | 0.99 | **0.96** | **0.98** | **0.01** | **0.00** | Global | 1.2M | This Study |
| PhaseNet | −0.02 | **0.11** | 0.96 | 0.93 | 0.94 | 0.09 | 0.01 | North California | 780K | 8 |
| GPD | 0.03 | 0.14 | 0.81 | 0.83 | 0.82 | 0.10 | 0.01 | South California | 4.5M | 10 |
| PickNet | 0.08 | 0.17 | 0.75 | 0.75 | 0.75 | 0.10 | 0.03 | Japan | 740K | 2 |
| PpkNet | 0.02 | 0.15 | **1.00** | 0.91 | 0.95 | 0.10 | 1.85 | Japan | 30K | 5 |
| Yews | −0.02 | 0.13 | 0.83 | 0.55 | 0.66 | 0.11 | 0.01 | Taiwan | 1.4M | 4 |
| Kurtosis | −0.10 | 0.13 | 0.89 | 0.39 | 0.55 | 0.11 | 0.01 | — | — | 17 |
| FilterPicker | −0.05 | 0.13 | 0.61 | 0.41 | 0.49 | 0.10 | 0.01 | — | — | 18 |
| AIC | −0.07 | 0.15 | 0.87 | 0.51 | 0.64 | 0.12 | 0.02 | — | — | 19 |

μ and σ are mean and standard deviation of errors (ground truth—prediction) in seconds respectively. Pr, Re, and F1 are precision, recall, and F1-score respectively. MAE and MAPE are mean absolute error and mean absolute percent error respectively.
Bold values represent the best performance.

earthquake. Here we used threshold values of 0.5, 0.3, and 0.3 for detection, P-picking, and S-picking respectively. A batch size of 500 and 30% overlapping is used during the pre-processing. We associated phase picks to individual events based on detection times. Hypoinverse[20] and HypoDD[21] are used to locate and relocate the associated events. Both travel time differences and cross-correlation were used for the relocation.

We detected and located 21,092 events within this time period (Fig. 5). This is more than a 2 fold increase in the number of events compared to Fukuyama et al.[22] who relocated 8521 events during the same time period with hand-picked phases provided by the JMA. Our catalog includes almost all of the events reported by JMA. We also note that our results were obtained using only a subset of the stations that were used by Fukuyama et al.[22] About 15 % of the associated events did not end up in the final catalog; however, this could be due to our simplistic association approach, and it is hard to assign them as false detections.

We used a local magnitude relationship[23] calibrated using reported magnitudes by JMA to estimated magnitudes of relocated events. The majority of newly detected and located events in our catalog are smaller earthquakes—with noisier waveforms- compared to those previously reported by JMA (Fig. 6a). We estimate the magnitudes of completeness (Mc) for JMA and our catalog as 1.82 and 1.50 respectively using the maximum curvature method[24]. While the frequency-magnitude distribution result (Fig. 6a) indicates that our deep-learning approach is effective in detecting and characterizing up to 20 times smaller microearthquakes, other factors such as better network coverage and smaller station spacing are required to decrease the overall magnitude of completeness[25,26].

In total, JMA's analysts picked 279,104 P and S arrival times on 57 stations, while EQTransformer was able to pick 401,566 P and S arrival time on 18 of those stations (due to unavailability of data for other stations). To compare the manual picks by JMA with

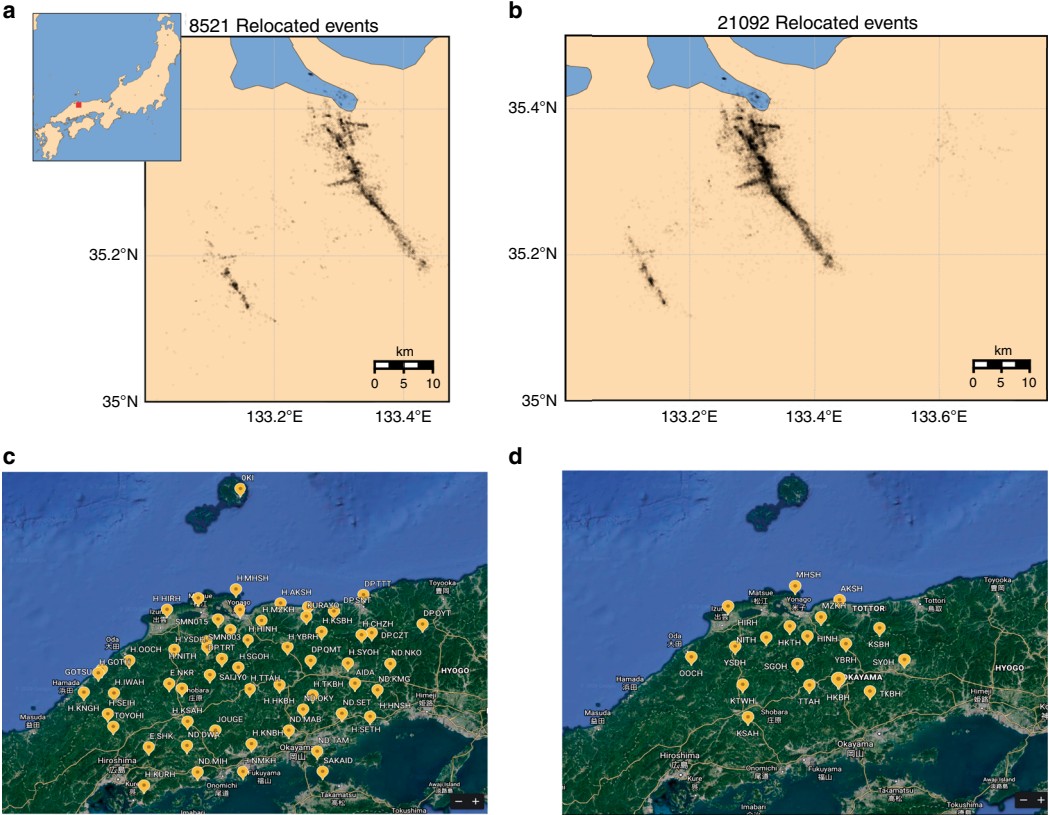

**Fig. 5 Tottori Earthquakes.** Seismicity of Tottori region between 6 October and 17 November 2000. **a** relocated events in Fukuyama et al.[22] using manual phase picks by JMA. **b** relocated events using the automatic phase picker (EQTransformer) of this study. **c** Distribution of 57 seismic stations used by JMA and Fukuyama et al.[22] **d** distribution of 18 stations used in our study to detect and locate earthquakes in Tottori region.

our automatic picks we used about 42,000 picks on the common stations and calculated the arrival time differences. The distributions of these arrival time differences between the manual and deep-learning picks for P and S waves are shown in Fig. 6b. The standard deviation of differences between picks are around 0.08 second with mean absolute error of around 0.06 second or 6 samples. Results are slightly better for S picks. The mean error is only 1 sample (0.01 s).

## Discussion

The better performance of the proposed method for phase picking could be due to several factors (e.g., quality and quantity of training set, architecture design, attention mechanism, depth of the network, the augmentations used during training process, etc). The attention mechanism helps to incorporate global and local scale features within the full waveform. A deeper network might result in more discriminatory power through learning of a more nonlinear mapping function.

Based on the test set results for our picker, errors seem to correlate with noise level (Supplementary Fig. 11). A similar correlation is seen between the variations in the predictionsand background noise level (Supplementary Fig. 12). We did not find a clear correlation between the estimated epistemic uncertainties (variations in the output probabilities) and picking errors. Aleatory uncertainties might provide better estimates for picking confidence intervals; however, such estimation of aleatory uncertainty for classification tasks is difficult[27]. Even so, knowledge of epistemic uncertainties and variability in the output probabilities can be useful to reduce the false positive rate.

Supplementary Fig. 13 presents examples of cultural noise recorded in Western Texas resulting in a false positive. The impulsive nature and frequency range of these arrivals makes it hard to differentiate them from an earthquake wave especially when a short window around the arrival is used. This can result in predicting a high probability for P or S picks. However, detection probabilities based on longer windows exhibit a higher variations/ uncertainties that can be used to eliminate the false defections. Including a large variety of anthropogenic and atmospheric noise into a training set would be an effective way to reduce such false positives; however, reliable labeling of such noise is itself a challenging task. Incorporating the spectral characteristics of the waveforms during the training process[7] might be another solution.

Picking P waves tends to be more uncertain for waveforms recorded at larger epicentral distances (Supplementary Fig. 12). These higher uncertainties could be due to having fewer long-distance waveforms in the training set and the fact that P waves can be more difficult to pick when the first arrival is emergent or is the diving wave Pn. The 1.0 Hz high-pass filtering of the data can also contribute to difficulty in picking the initial onset. As expected, we observe higher uncertainties in picking smaller events (Supplementary Fig. 12). We also note that lower prediction probabilities exhibit a higher uncertainty level and that the model outputs lower probabilities for P-wave picks with lower SNR, larger event-station distance, or smaller magnitude. Such tendencies are not as strong for the S-picks (Supplementary Fig. 14).

The geographical location and the size of training data do not seem to be the main factor controlling performance. PhaseNet has very good performance even thought it was trained on data

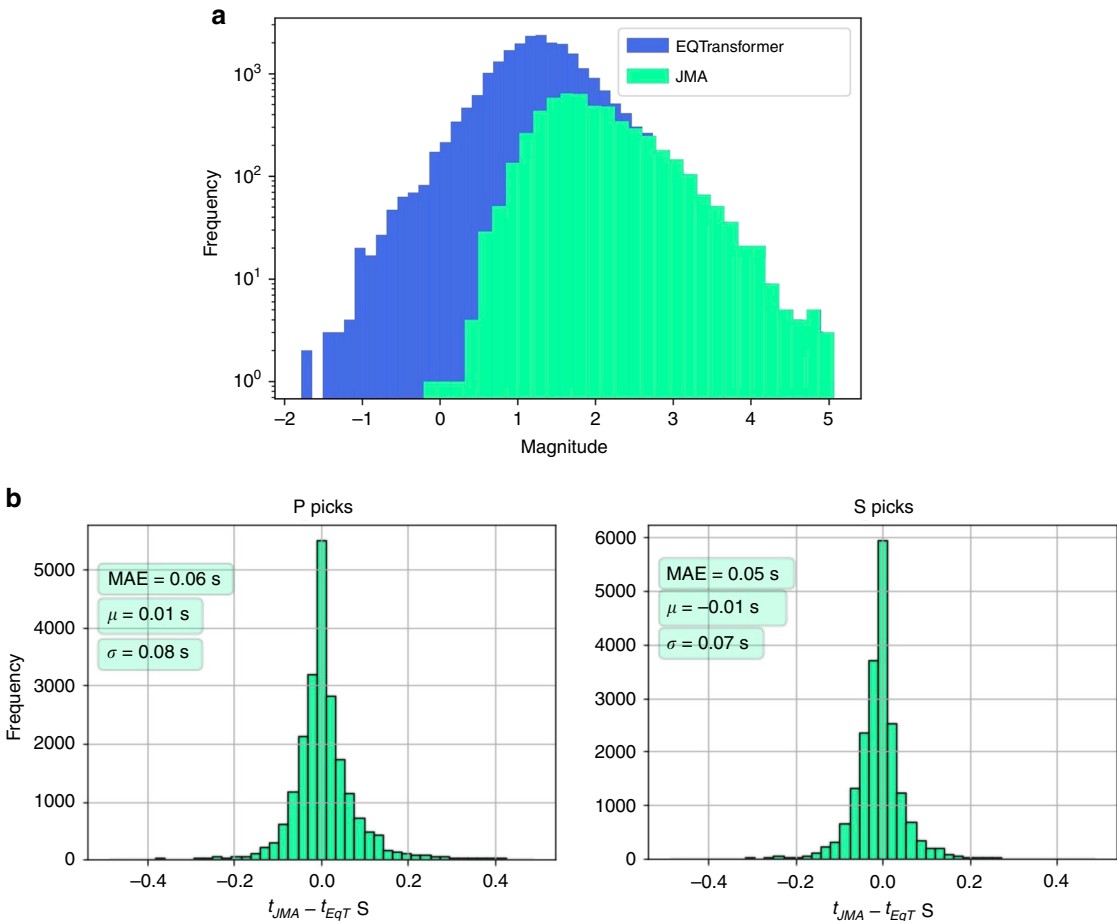

**Fig. 6 Distributions of frequency magnitude of earthquakes and picking errors. a** frequency-magnitude distributions of located events in JMA catalog and relocated events in our catalog (EQTransformer). Magnitudes for all events have been estimated using a local magnitude scale. **b** distributions of arrival-time differences (in second) between P (left) and S (right) picks by JMA's analysts and EQTransformer.

only from the Northern California. This indicates that deep-learning models trained by data set in specific region can generalize well to other regions and that deep-learning pickers for most purposes can be used off the shelf without retraining. PpkNet[5] that was trained by 30 K waveforms resulted in higher F-scores for both P and S waves compared to the other models that were built using much larger training sets. This suggests other factors such as the network type (e.g., recurrent vs convolutional), training process (e.g. the use of augmentation), or/and quality of a training set can play a more important role than the size of the training set.

The precision of picking seems to be influenced more by the labeling and training procedure. For instance, the sporadic error distributions for P-picks in the Yews[4], GPD[10], and PpkNet[5] results may be due to their training procedure that render them sensitive to arrival times clustered at particular time points. Compared to the traditional pickers, deep-learning-based methods perform better for noisier waveforms - especially for the S waves (Fig. 7).

Performing comparative analyses on models with different properties is a very challenging task. Different models have adopted different labeling approaches and incorporated different network designs. This results in different sets of hyperparameters that can affect the model performance extensively. Quality of a training set and the training procedure are other important factors that are hard to quantify and measure their influence. On the other hand, setting up a fair environment for the comparison and

utilizing a reliable and independent benchmark are important for a more unbiased assessment. Despite all these deficiencies, we hope to initiate such efforts and encourage our colleagues in the seismological community to pursue more rigorous testing and comparative analyses to learn from and build on previous attempts.

Traditional pickers do relatively well in the accuracy of arrival-time picking while their main disadvantage is generally lower recall and poorer performance in picking S phases (Tables 2 and 3). Non-symmetric error distributions of traditional pickers (Supplementary Figs. 9 and 10) are primarily due to skew introduced by their systematic delay in picking the arrival times, which is more significant for S waves; however, their comparable performance to some of the deep-learning models indicates their effectiveness even though they do not require training. We also note that the traditional pickers are not necessarily faster. For instance, on a machine with a 2.7 GHz Intel Core i7 processor and 16 GB of memory it takes 62 hr and 12 min, 3 hr and 25 min, and 31 hr and 18 min for Kurtosis, FilterPicker, and AIC pickers (based on the python implementation in PhasePApy[28]) respectively to pick the entire test set, while EQTransformer finished the detection/picking in 2 hr and 28 min (on the same machine).

Our applications of EQTransformer to Japanese data indicates high generalization and accuracy of the model. The precision of arrival time picks by EQTransformer are comparable to manual picks, and its higher sensitivity results in more than twice the number of detected events. The newly detected events are not

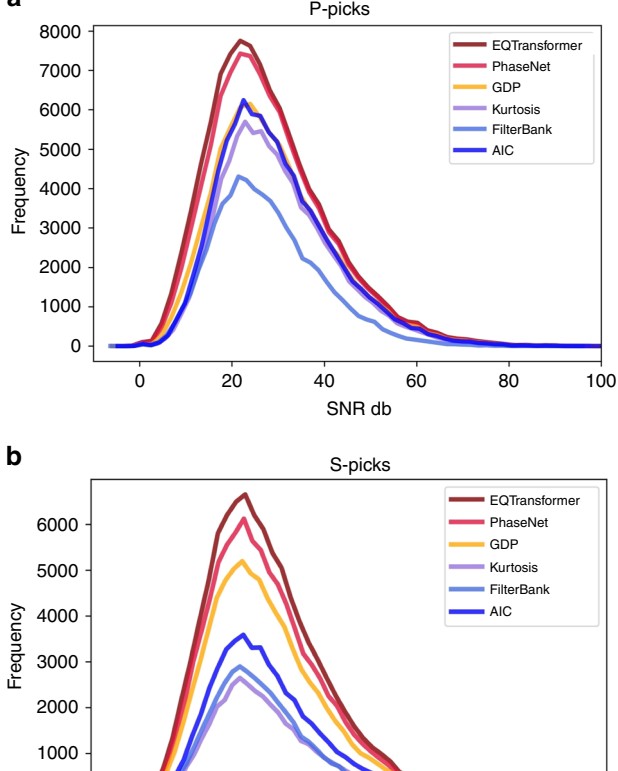

**Fig. 7 Phase picking performance as a function of noise level.** P (**a**) and S (**b**) phase picks as a function of signal-to-noise ratio (SNR) for three deep-learning and three traditional pickers.

limited to the two main faults active in the Tottori sequence, and include sparse seismicity under the eastern flank of Mt. Daisen. This result was attained using only a portion (less than 1/3) of the seismic stations, with relatively large station spacing, and a simple association method h. Using more advanced phase association algorithms (e.g., Glass3[29]) would likely further increase the number of events. The effectiveness of the method together with its high efficiency (the processing time for 1 month of continuous data at 1 station is 23 minutes on a single CPU without uncertainty estimation) highlight the potential of the proposed method for improving the earthquake monitoring.

In this paper, we present a multi-task network for simultaneous earthquake detection and phase picking based on a hierarchical attentive model. Our network consists of one deep encoder and three separate decoders. Two levels of self-attention (global and local) are embedded into the network in a hierarchical structure that helps the neural network capture and exploit dependencies between local and global features within an earthquake waveform. Our model has several distinctive characteristics: (1) it is the first hierarchical-attentive model of earthquake signal; (2) with 56 activation layers, it is the deepest network that has been trained for seismic signal processing; (3) it has a multi-task architecture that simultaneously performs the detection and phase picking while modeling the dependency of these tasks on each other through a hierarchical structure; (4) in addition to the prediction probabilities, it provides output variations based on Bayesian inference; (5) it is the first model trained using a globally distributed training set of 1.2 M local earthquake observations.

## Methods

**Related work.** Perol et al.[30] used a network of 8 convolutional and one fully connected layers to detect and cluster events based on three component waveforms. They trained their network using ~2700 earthquake and ~700,00 noise waveform recorded in Oklahoma and tested it on 209 events and ~131,000 noise waveform from the same region. They mainly compared their approach with similarity search methods and concluded that a deep neural network can achieve superior performance in less computational time. Wu et al.[31] applied a densely connected network of 7 fully convolutional layers to detect laboratory earthquakes (1000 samples) of different sizes. Ross et al.[10] trained a network of 4 convolutional and 2 fully connected layers using 4.5 Million seismograms recorded in Southern California to detect short windows of P-waves, S-waves, and noise. They applied the trained network to 24 h of continuous data recorded at a single station in Bombay Beach, California and a single event recorded by multiple stations in Japan and showed that deep neural networks are capable of detecting events with different waveform shapes than those used for the training without sacrificing detection sensitivity. Ross et al.[9] adopted a similar approach (3 convolutional and 2 fully connected layers) for picking P arrival times. Zhu and Beroza[8] used U-Net, a fully convolutional encoder-decoder network with skip connections, for an end-to-end picking of P and S phases. They trained their network using 780 K seismograms and tested it using 78 K seismograms recorded in Northern California. Mousavi et al.[7] proposed a residual network of convolutional, bidirectional Long Short Term Memory units, and fully connected layers for detecting earthquake signals in the time-frequency domain. They used 500 K 3-component records of noise and tectonic earthquakes from North California for training the network and tested the performance of the final model by applying it to semi-synthetic data and one month of continuous seismograms recorded during a sequence of induced seismicity in Central Arkansas. This study showed deep-learning-based models can generalize well to seismic events with substantially different characteristics that are recorded in different tectonic settings and still achieve high precision even in the presence of high background noise levels with low computational cost. Pardo et al.[6] also used ~774 K seismograms form Northern California to train their two-stage phase picker. They used a convolutional network for a rough segmentation of phases first, and then in a second stage performed a regression to pick the arrival times. Zhou et al.[5] (~136 K augmented P and S waveforms) and Zhu et al.[4] (~30 K) used seismic data from the 2007 Wenchuan aftershock sequence in Sichuan, China to train deep-leaning -based detectors and pickers. While Zhou et al.[5] used two separate networks of an 8-layer convolutional networks and a two-layers of bidirectional Gated Recurrent Units for detection and picking respectively, Zhu et al.[4] used the same network (11 convolutional and 1 fully connected layers) in a recursive manner for both detection and at the cost of larger computational time. Dokht et al.[3] trained two separate networks each consisting of 4 convolutional and 2 fully connected layers for detection and a rough estimation of P and S arrival times in the time-frequency domain. They used ~162 K waveforms recorded in Western Canada for training. Wang et al.[2] built two separate models based on modification of VGG-16 network and ~740 K seismograms recorded in Japan for picking P and S arrival times respectively. Their models work for short time windows that are roughly centered around the S phase arrival. This centering is done using the theoretical arrival times, which in practice are unknown without information about the earthquake locations.

These studies not only differ in network architecture and overall approach, they also employ different data pre-processing, augmentation techniques, use datasets of different sizes, magnitude range, epicentral distances, event types, noise levels, geographical locations, and report the results using different matrices (e.g. accuracy, precision, recall, F1-score, average precision, hit rates, absolute error, picking error, etc.) that make it difficult to determine the relative performance, strengths, and weaknesses of each method. This prevents the community from adopting and building on the most effective approach. This is due in part to the lack of a standard benchmark dataset with high quality labels to facilitate rigorous comparisons. The data set used in this study (STanford EArthquake Dateset[13]) is a candidate standard benchmark for developing and comparing detection and phase picking algorithms for local earthquakes.

**Network design.** Seismic signals are sequential time series consisting of different local (individual seismic phases) and more global (e.g. packages of body and surface waves and scattered waves) features. Hence, it is useful to retain the complex interaction between the local and global dependencies in an end-to-end deep learning model of seismic signals. Traditionally, recurrent neural networks have been used for such sequence modeling; however, relatively long duration seismic signals require some down sampling prior to the recurrent layers to manage the computational complexity. Hence, a combination of recurrent and convolutional layers has been shown to be an effective architecture for sequential modeling of seismic signals[7]. Building upon our previous work[7], we introduce a multi-task network of recurrent and convolutional layers that incorporates attention mechanism as well. Attention mechanism is a method of encoding sequence data in which elements within a sequence will be highlighted or down-weighted based on their importance or irrelevance to a task[32–35]. The overall structure of our network includes one encoder and three separate decoders. The encoder consumes the seismic signal in the time domain and generates a high-level representation and

contextual information on their temporal dependencies. Decoders then use this information to map the high-level features to three sequences of probabilities associated with: existence of an earthquake signal, a P-phase, and an S-phase respectively.

**Very deep encoder**. Several studies[36–38] have shown in end-to-end learning from raw waveforms that employing deeper networks can be beneficial for having more expressive power, better generalization, and more robustness to noise in the waveform. We build a very deep encoder, which is known to be important in performance of a sequence-to-sequence model with attention.

In self-attentive models the amount of memory grows with the sequence length. Hence, we add a down sampling section composed of convolutional and max-pooling layers to the front end of the encoder. The encoder follows with several blocks of residual convolution layers and recurrent blocks including network-in-network connections.

Convolutional layers exploit local structure and provide the model with better temporal invariance, thus typically yielding a better generalization. To be able to extend the depth of the network without degradation we use blocks of convolutional layers with residual connections[39] as depicted in[39] (Supplementary Fig. 15).

Long-short term memory (LSTM)[40] are specific types of recurrent neural networks commonly used for modeling longer sequences. The main element in a LSTM unit is a memory cell. At each time step, an LSTM unit receives an input, outputs a hidden state, and updates the memory cell based on a gate mechanism. Here we expand the bidirectional LSTM blocks by including Network-in-Network[41] modules in each block to help increase the network's depth without increasing the number of learnable parameters (Supplementary Fig. 16). LSTM layers prior to self-attentional layers, have been shown to be necessary for incorporating positional information[42–44].

**Attention mechanism**. We represent the output of an LSTM layer by $H = \{h_t\} \in \mathbb{R}^{n \times d_h}$ as a sequence of vector elements (a high level representation of the original input signal), where, $n$ is the sequence length and $d_h$ is the dimension of the representation. We calculate the self (internal) attention as follows[45,46]:

$$e_{t,t'} = \sigma(W_2^T[\tan h(W_1^T h_t + W_1^T h_{t'} + b_1)] + b_2) , \qquad (1)$$

$$\alpha_{t,t'} = \frac{exp(e_{t,t'})}{\sum_{t'} exp(e_{t,t'})} , \qquad (2)$$

$$c_t = \sum_{t'=1}^{d_h} \alpha_{t,t'}.h_{t'} , \qquad (3)$$

where $h_t$ and $h_{t'}$ are hidden state representations at time steps $t$ and $t'$ respectively. $W$ and $b$ are weights matrices and bias vectors respectively. $\sigma$ is the element-wise sigmoid function. $\alpha_{t,t'}$ are scalar scores (also called alignment) indicating pairwise similarities between the elements of the sequence. The attentive hidden state representation, $c_t$, at time step $t$ is given by summation of hidden states at all other time steps, $h_{t'}$, weighted by their similarities to the current hidden state, $\alpha_{t,t'}$.

Vector $c_t \in \mathbb{R}^{d_h}$ is a sequence of context-aware (with respect to surrounding elements) encoding that defines how much attention will be given to the features at each time step based on their neighborhood context. This will be incorporated into a downstream task as an additional contextual information to direct the focus to the important parts of the sequence and discard the irrelevant parts.

We adopt the residual attention blocks introduced in the Transformer[47,48]. We replace the multi-head scaled-dot product attention by above single-head additive attention (Supplementary Fig. 17). The feed-forward layer consists of two linear transformations with a ReLU activation in between, $FF(x) = \max(0, xW_1 + b_1)W_2 + b_2$, intended to introduce additional nonlinearities.

Our goal is to implement two levels of attention mechanisms in a hierarchical structure[46,49,50] at both the earthquake full waveform, and individual phase levels. A logical way to do this is to perform attention mechanisms at two levels with different temporal resolutionsuch as: applying the detection attention at the high level representation at the end of the encoder and the phase attention at the end of associated decoders where higher temporal resolutions are available. With $O(n^2.d)$ complexity of self-attention, however, this is not computationally feasible for the long duration time series (6000 samples) used here. Hence, we applied attention mechanisms with both global and local attentions at the bottleneck. The attention block at the end of the encoder performs global attention, by attending to all the positions in the sequence, to learn to identify the earthquake signals within the input time series. The shortened path from this layer to the detection encoder and the naturally higher detection loss make this learning easier.

Attention blocks at the beginning of phase-picker decoders perform additional local attention by attending only to a small subset of the sequence[45]—aiming to sharpen the focus to individual seismic phases within the earthquake waveform. One LSTM layer with 16 units is applied before the first attention block at each level to provide position information[42,51].

**Uncertainty estimation**. Model uncertainty is important in applied deep-learning, and for seismic monitoring; however, none of the previous deep-learning model for earthquake detection/phase picking provides a probabilistic output with a measure of model uncertainty. The predictive probabilities provided by these models are not equivalent to model confidence. A model can be uncertain in its predictions even with a high softmax output[52].

In deep learning, the model uncertainty is usually estimated by inferring distributions over the network weights. Due to the computational complexity, this is done by approximating the model posterior using inference techniques. Gal and Ghahramani[52] showed that dropout[53], a technique commonly used to prevent overfitting, can be used for approximating Bayesian inference over the network's weights. This is done by using the dropout at test time to impose a Bernoulli distribution over the network's weights. This is equivalent to Monte Carlo sampling from the posterior distribution over models. We implement a dropout after every layer of our neural network and use it during both training and prediction.

## Data availability

STanford EArthquake Dataset (STEAD) used for the training, validation, and test is available at: https://github.com/smousavi05/STEAD. The continuous data for the Totorri region was downloaded from HiNet (http://www.hinet.bosai.go.jp/about_data/?LANG=en). Maps and figures in this paper were generated using the Generic Mapping Tools and Matplotlib[54]. Catalog of all events detected and relocated in our study are provided in the supplementary materials. Source data are provided with this paper.

## Code availability

Our source code and model are available at https://github.com/smousavi05/EQTransformer and can be used to apply the model to continuous data or reproduce results presented in the paper.

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

## Acknowledgements

We would like to thank Yijian Zhou for testing PpkNet models on our test set. Eiichi Fukuyama, Kaz Imanishi, Shin Aoi, Takanori Matsuzawa helped us to acquire HiNet data for the Tottori sequence. S.M.M. was supported by the Stanford Center for Induced and Triggered Seismicity (SCITS) and G.C.B. was supported by AFRL under contract number FA9453-19-C-0073.

## Author contributions

S.M.M. designed the study, implemented the method, performed the tests and conducted the analyses. W.L.E. designed the Tottori test and helped with the interpretation of analyses results. W.Z. applied the PhaseNet on test set. L.C. applied Yews on the test set and performed some analyses. G.C.B. lead the project, and reviewed the manuscript. S.M.M. and G.C.B. wrote the manuscript. All authors discussed extensively the results.

## Competing interests

The authors declare no competing interests.
