## [Peer Review File · Nature Communications]

REVIEWER COMMENTS

Reviewer #1 (Remarks to the Author):

Earthquake Transformer: An Attentive Deep-learning Model for Simultaneous Earthquake Detection and Phase Picking

Summary: A new/updated technique is presented for earthquake detection and phase arrivals extracted from those detections; notably the method applies an attention layer to the CRED model to increase performance using 3 parallel detectors. This is a needed study and I would recommend for publication pending revisions to address the comments presented.

The paper provides an analysis of seismic event and phase detection, which from the comparisons presented is the subject of multiple recent publications. The technique performs well but there are some details missing in the reporting that would help the reader assess the performance.

(1) All detections presented require a threshold but nothing is mentioned and the information is in the supplement. See comments below.

(2) The comparison of other studies is well needed. Please be more concise on exactly what is presented for the model comparison statistics. Are the values from the text? Or are the models available and used 'as is' to evaluate? If pretrained models are available, using the STEAD data to test would be great, however that would require manipulating the input data for each model design.

(3) The discussion section wanders around and vaguely refers to supplement material leaving the reader to figure out exactly what is presented. Consider revising.

(4) The Japan analysis is good but add details. See below.

Major comments:

p10. "The shorter path through the detection decoder and its higher loss (due to longer length of label) naturally force the network to first learn to distinguish the earthquake signal within a time series." What does this refer to? The decoder layers are trained in parallel if my understanding is correct.

p11. Confusion matrix. 1 FP and 1-FN is impressive. What is used to determine a TP and TN? The softmax output will require a threshold to count as a TP; from supplement figure 4 the value is probably around 0.6. That said, every example shown is a detection softmax of ~ 1 . Are there no situations where non earthquake signals appearing on all 3 channels produce a softmax prob of ~ 0.5 , eg. the waveforms in Fig 4f occurs all the time in continuous records from non-tectonic signals. The noise used in model training is 200k samples and the earthquakes are 450k. How diverse is the noise? I suspect the model will show less confidence in the softmax predictions if tested with variable noise containing energy in the 1-15 Hz band, which is found in many environments.

Table 1. It is interesting that DetNet and Yews underperform to the classic averaging technique. In Yews (Zhu et al., 2019) they train with Wenchuan data only. Are the results here presented here using pretrained models from the authors and tested with STEAD? If so make that clear to show the other models do not generalize to seismic waveforms outside the study area. Or are the statistics taken from these studies.

The discussion mentions the performance of Yews, GPD, and PkpNet is decreased because of windowing. Here, 1 minute windows are used. For continuous data how does the model handle events that occur near the edge of a 1-minute interval? How does the model handle a detection

that is split by a window? How does the model handle small events occurring within seconds, i.e. multiple in 1 time window?

p15. It is difficult to assess why the p-waves are more uncertain than the s-waves. Supp. Fig 4 shows p and s waves with precision of 1 at all thresholds but a recall less than 1 for s wave threshold of 0.3 compared to 0.6 for p waves. A clearer presentation of how the detection threshold metrics are applied and reported is needed.

p16. It is not clear why the suggestion of using traditional pickers to improve the CNN picks is presented when Figure 7 shows EQTransformer having better statistics.

The independent Japan analysis is good but details are lacking that would improve clarity. Per comments above: How does the model handle events occurring between window intervals? What threshold is set to determine a detection? Does cultural and environmental noise show increased softmax probabilities? How many false detections are removed by hypoinverse in the association of events? Is the increase in $M > 0$ events due to JMA staffing constraints that cannot spend time hand picking the smallest events? Overall the performance is good but 2-3 sentences of details would improve.

Minor comments:

All figure captions need more detail for the reader to understand the information presented.

Figure 2 caption; check typo.

Figure 3. Very nice visual description for the attention layer but add axis labels and convert x-axis to time (s), not index position.

What is the right axis? The summation of the weighted hidden states?

Figure 4. Add detail to what is exactly shown. Multiple channels I presume. Why is the predicted shown on one channel and manual picks shown on 2 channels? State what the confidence interval shown in the model prediction panel. It is not obvious it is there until 4f. Show the meta data for each event, e.g. magnitude and distance, to give the reader information of what is shown. Label each axis.

Christopher Johnson

Reviewer #2 (Remarks to the Author):

Review of the manuscript:

"Earthquake Transformer: An Attentive Deep-learning Model for Simultaneous Earthquake Detection and Phase Picking" by Dr Mousavi and colleagues,

The article provides a novel deep learning methodology to detect earthquakes and pick P- and S-wave onset phases. The technique is applied to the STEAD dataset and to the continuous data recorded in the aftershock region of the 2000 Mw 6.6 western Tottori earthquake. The results presented are remarkable and of great interest to a wide group of seismologists spanning from researchers in tectonics (larger number of located earthquakes resulting from lower magnitude detection thresholds) to operational seismic monitoring centers in charge of real-time earthquake detection and the compilation of national to global earthquake catalogs.

Overall, the adoption of the recurrent/transform/attentive scheme is novel and it appears to make a step forward towards an improved adoption of ML techniques in seismology. Because of the complexity of the methodology (several interconnected stages) and of the use of state-of-the-art

solutions, the method section requires a solid background in the machine learning literature. Nonetheless, the addition of some detail on the choices made to design the methodology would be likely help the reader to follow and possibly replicate the proposed model and reproduce the results.

This is an important manuscript that should be published by Nature Communications. I recommend publication after minor revision.

Alberto Michelini (with the collaboration of Dario Jozinović)

General advice

It would be a good practice that throughout the article whenever there is made reference of the supplementary material it is also indicated what exactly therein (e.g., Fig. X of supplementary material, Supplementary Note Y, ...).

Main comments and questions

1. Additional details should be provided in the Training section. This should include the criteria used to select and optimize the network architecture and hyperparameters, i.e. how did the authors select the number of layers, the kernel size for the convolutional layers etc.
2. It would be interesting to show some waveforms examples in which the EQTransformer performs badly (which seems rare), to try to understand if there are any, and what are the limitations of EQTransformer.
3. Related to 2., it is not clear how the algorithm addresses cases in which there are more than 1 earthquake within the 60 s window. More specifically, since the training was done using the STEAD dataset which was constructed adopting only one (1) earthquake within the 60 s time window as criterion (cf 1st paragraph of the Earthquake Waveforms section of Mousavi et al., 2019). I have the following questions
 - 3.1. How were the time windows applied and shifted through the continuous data ? If overlapping windows were used, what was the shift in seconds ?
 - 3.2. What occurs when the window includes two or more earthquakes with the same 60 s window ? This is an important issue since the ability to detect and pick P and S phases in the early minutes of an aftershock sequence is crucial.
 - 3.3. At p. 10 there is provide a nice description of how the attention mechanism works and it would be important to see what occurs with the attention mechanism when two earthquakes are in the same window (i.e., the Fig. 3 with two earthquakes within the 60 s window). It would be important to add this special case amongst the different types of representative waveforms of Fig. 4.

Other comments and questions

1 Introduction

P 2 bottom

"...0.01 second of error in determining P-wave arrivals can translate to hundreds of meters of error in location."

This seems an exaggeration. Assuming an average $V_p = 6.0$ km/s, 0.01 s would result in no more than 60 m. Rather, there are other factors that can be more relevant including e.g. inclusion or exclusion of P and S phases which, when using not well calibrated local velocity models, can result into large changes of the earthquake focus.

P3 mid

The authors describe the basic constituents of the EQTransformer model and it can be helpful for

the reader to have right away a reference to the basic literature that regards the attentive-based models (ie.g., ref. 47, Vaswani, et. al., 2017).

2 Related work

P 5. Top

"... They trained and tested their network using earthquake and noise data recorded in Oklahoma (~2700 events and ~700,00 noise windows) and ..."

The test data set numbers used by Perol et al. (2018) are missing.

The following is what Perol et al. report in their paper

The test set contains all the windows for July 2014 (209 events and 131,072 windows of noise), whereas the training set contains the remaining windows (2709 events and 700,039 noise windows).

3 Results

Data and Labeling

P 8 First paragraph:

Description is misspelled

P 9 top

"...band-passed filtered between 1-45 Hz."

What is the basis for choosing these values ? The STEAD data set contains also broadband data and the high pass corner (1.0 Hz) seems a little too high and it may be one of the reasons why the methodology performs more poorly at larger epicentral distances.

P9 top

"...Triangular labeling ... adopted for the final model. In this form, probabilities of P and S are set to 1 right at the first arrival time and linearly decrease to 0 within 20 samples before and 20 samples after. "

Does this imply that effectively the same weight is given to the P phases regardless of the picking quality ?

Training

See main comment #1

Comparison With Other Methods

P11 first paragraph

"This can be seen clearly from the example in Figure 4-c, where two strong and apparently separate waves are detected as parts of a single event rather than two individual events."

If the training was performed on a data set that featured only one earthquake per time window, is it possible that the model can detect two earthquakes in the same window ? Please clarify since this answers also the previous questions on multiple earthquakes in the same time window (main comment #3).

Tables 2, 3

Please verify the values for MAE (and MAPE).

According to the definition given in the supplementary material and assuming that the true values consists of arrival times reported in STEAD and the predicted are those resulting from the different models, the values of MAE appear too large. For example, if we consider e.g. 10000 sample errors with 0 zero mean and a standard deviation sigma of 0.3, the resulting value of MAE would be ~0.024.

In contrast, the MAE values for P and S picks of Figure 10 are perfectly consistent with the standard deviations provided.

Regarding the MAPE, since the true value ($t_{\text{catalogue}}$ in figs 6 and 7) appear at the denominator (see supplement), it is not clear how MAPE is determined.

Did the authors take the travel time as the true value ?

(The MAPE values are not provided for the Tottori earthquake sequence)

P12

"A pick was considered as a true positive when its absolute distance from the ground truth is less than 0.5 second"

It would be useful to evidence this information also in the captions of Fig. 6 and Fig. 7.

P12 bottom

"...The error distributions for some of the deep-learning pickers are not uniform and are not uniform and cluster at sporadic times due to their moving-window scheme. "

Can the authors elaborate a little more on this ?

P14 end of the section

"Results are slightly better for S picks."

This somewhat slight improved performance of the S wave picks observed for the Tottori earthquake sequence is intriguing. It is well known that S-waves onsets are much more difficult to pick and it is common to observe significantly larger std for S waves. This behavior is also observed with the STEAD data set and shown in Figures 6 and 7 for the EQTransformer and the other methodologies. Can the authors articulate their thinking on this observation and reconcile with the results obtained using STEAD ?

Discussion

This section discusses the results using figures in the supplementary material. For completeness of the main text, I would advise the authors to insert the figure(s) they consider the most significant into the main text.

P14 1st paragraph

"A deeper network can result in more discriminatory power"

Correct but it is not known yet. It would be preferable to use "might" rather than "can".

P14 2nd paragraph

"errors seem to correlate with noise level (supplementary information)"

There are several figures in the supplementary information regarding the errors and their variation according SNR, distance and magnitude. Which ones are the authors referring to ?

P14 bottom paragraph

"We did not find a clear correlation between the estimated epistemic uncertainties and picking errors."

Can you articulate on what you consider the picking epistemic uncertainties

P15 top

"...and the fact that P waves can be more difficult to pick when the first arrival is the diving wave P_n ."

This difficulty can also originate from the 1.0 Hz highpass filtering adopted. This should be mentioned.

P15 last paragraph

"...PickNet may have a similar problem since the training is done based..."

Figure 6 does not seem to show the problem described for the other ML automatic pickers. It may

not be proper to mention it.

P16 top

"Here we used estimates of theoretical arrival times to center the input windows around each phase as was suggested in..."

In this study, this approach was used only when training and not when testing. Correct ?

P16 last paragraph

"These comparisons suggest there may be advantages to hybrid approaches in which deep-learning-based picks are refined either by applying a traditional picker or cross-correlation delay measurements to improve the results."

It is not entirely clear what the authors want to say. Anyway, if the objective is to obtain cross-correlation picks, it may be just necessary to identify the P (and S) windows to be cross-correlated with a target waveform and it may not be needed the whole procedure described in this article which also seem to be very effective already.

P17 top

"...and a simple association method h. Using more advanced phase association algorithms (e.g. Glass3)..."

Would it be possible to have more information on the simple association method ? How many phases that were picked by EQTransformer were not associated to an earthquake ?

Methods

Very deep encoder

"We build a very deep encoder, which is known to be important in performance of a sequence-to-sequence model with attention. "

Related to main comment #1 and more specific, have the authors tried with a not as deep encoder ? It would be important to know if within the same scheme a deep encoder is really needed.

Attention Mechanism

P19

The definitions of n and d_h are missing.

P20

"We replace the multi-head scaled-dot product attention by above single-head additive attention Figure 14."

Can you explain why you replace it ?

Supplementary material

Note 2. I did not find the Catalog of all events detected in the material downloaded

Supplementary movies. Probably just one movie is enough. The focus of the article is not the Tottori earthquake sequence but the new EQTransformer technique.

Figure 2 (or 3). Please add case with two or more earthquakes within the same 60 s window.

Figure 5. Are there used two different color scales for P and S absolute errors ? Is the color proportional to the number of counts for each pixel ?

Reviewer #3 (Remarks to the Author):

This manuscript documents a novel deep-learning approach for local/regional earthquake detection and P and S wave picking. The approach uses an attentive model which allows it to focus on local features (P and S wave arrivals) to improve upon arrival-time picking following detection. The model is unique in that it is trained using data from a set of global earthquakes. The manuscript does an extensive job of comparing the performance of the model as compared to both traditional pickers (e.g., sta/lta) and previously published deep-learning approaches. The approach is further validated by testing its performance on detecting and picking a sequence of earthquakes in Japan. I've attached a marked-up document with the majority of my comments.

The authors do a good job of describing the model and it is clear that this represents the cutting edge of the application of deep-learning to earthquake detection and picking. The comparison of this model with other methods is above and beyond what is typically done and emphasizes the improved performance of the model architecture/training employed. Overall, I have few comments in regard to the work presented here, although I would have liked to see more description on how it is applied to continuous data.

I think the presentation, organization, and resulting impact of the manuscript could be improved. The manuscript is written expecting a fairly high level of understanding of machine learning, and to most seismologists, I expect it will not be clear how this is a significant step forward in the application of these technologies to earthquake monitoring.

In terms of organization, there is a large portion of the introduction that details previous works, highlighting the distinct architectures of each approach. I think this level of detail will go beyond most readers and distracts from highlighting what makes this approach unique. Some of the features highlighted in the model (in the six main points) will also go beyond most readers and in themselves are not as important as the improved performance. (e.g., why does it matter that this is the deepest model? why does it matter that this is the first attentive model? why does it matter that it combines recursive and CNN architectures?).

The results section discusses the architecture and training of the model, but I think this should be placed entirely in the methods section. Instead, I think the results should immediately highlight the improvements of the model as compared to previous methods and other important features like its huge leap in computational efficiency. The application to continuous data is important and I don't think it is highlighted as it should be. If the authors could point to what new observations come from creating such a high-quality dataset and how efficiently they are able to produce it, I think it would strengthen the impact of the paper significantly.

I am a little concerned over the comparisons to other methods, but I am thrilled to see the attempt. It's a difficult problem because each deep-learning model was trained with distinct applications in mind and because more general methods can be optimized for specific use cases. I think the authors should expand upon the criteria used for the different standard pickers (filtering, STA/LTA windows, etc.). The testing set used to validate the models is a subset of the overall data set. Therefore, it was selected following the same criteria as the training data. This means that there could inherently be some bias that would improve the apparent performance of their model.

Some quick points/questions not in the PDF:

What happens when you have signals from events beyond 300 km (e.g., teleseisms?)

From supplemental figure 5, it looks like your data is dominated by events within 100 km? Why are there so few records beyond 100 km?

Somewhere in the figure/table captions, you should mention MAE is in samples as opposed to seconds.

I don't think the movies are necessary at the moment because the tectonic interpretation is not highlighted in the paper.

Long term, have you considered trying to isolate Pg vs Pn, Sg vs Sn at these regional distances?

REVIEWER COMMENTS

Reviewer #1 (Remarks to the Author):

Earthquake Transformer: An Attentive Deep-learning Model for Simultaneous Earthquake Detection and Phase Picking

Summary: A new/updated technique is presented for earthquake detection and phase arrivals extracted from those detections; notably the method applies an attention layer to the CRED model to increase performance using 3 parallel detectors. This is a needed study and I would recommend for publication pending revisions to address the comments presented.

thank you for your comments and suggestions. I provide point by point responses in the following and try to address all of your comments and suggestions. However, I should point out the presented networks has substantial differences to CRED in both form and intend. CRED is a detector designed of convolutional, recurrent, and fully connected layers performing in time-frequency domain. EqTransformer, is a multi-task network that does both detection and phase picking in the time domain that beside the attention mechanism and network-in-network layers even in the use of convolutional and recurrent layers differs significantly from our previous work (CRED)

The paper provides an analysis of seismic event and phase detection, which from the comparisons presented is the subject of multiple recent publications. The technique performs well but there are some details missing in the reporting that would help the reader assess the performance.

(1) All detections presented require a threshold but nothing is mentioned and the information is in the supplement. See comments below.

The optimal threshold value (that minimizes both false negative and false positive rates) differs from model to model and is the subject of characteristics of each model and the dataset of interest. For each model we tested a few threshold values based on recommendations of its authors and the best performance (minimum false negative and false positive rates) were reported at the end. In total 13 different models (for detection and picking) have been used here for the comparison and some of these models have other parameters like window size, overlapping, shift, etc in addition to the threshold values. For instance Kurtosis has about 11 parameters. There is long list of hyperparameters used for the location and relocation process as well. Reporting all of these parameters and in the main text will add too much unnecessary details that won't help comparing the performances. The important fact is that we (and the authors who helped us to test their models) reported their best performance. However, I added two tables into the supplementary material section listing some the most important parameters.

(2) The comparison of other studies is well needed. Please be more concise on exactly what is presented for the model comparison statistics. Are the values from the text? Or are the models available and used 'as is' to evaluate? If pretrained models are available, using the STEAD data to test would be great, however that would require manipulating the input data for each model design.

As you pointed in your earlier comment and we also mentioned in the manuscript detection and phase picking have been the subject of many studies and many papers have been published so far. Despite of all these works and effort, we still do not now the answer to some of our basic questions such as which network type (convolutional, recurrent, fully connected, or a combination) is the most suitable one for these tasks and earthquake signals? Is it better to approach the detection and picking separately (either using different network of a same one) or a hybrid method would be more appropriate? Should we perform these tasks in time or time-frequency domain? what should be the standard preprocessing procedure? what augmentation techniques are needed? what is the ultimate power of deep-learning models? are they like a high-level template matching that need to be train for each region (as Perol el at suggests) or no they can learn something beyond similarity of waveform pairs? etc. These are such questions that have been answered in each specific sub-field of machine-learning like vision, speech, and NLP. These un-answered questions preventing us from moving forward to practical applications in routine earthquake monitoring. A standard benchmark dataset with high-quality labels, and performance of baseline methods is only the first step toward this goal that hopefully eventually bring us closer to the answer of some these questions.

This has not been done before: 1) because it is extremely time consuming. As an example published models have been designed based on different waveform length? they have different hyper parameters? used different platforms? etc. It took between 1 week to 2 months to produce each subplots in figure 7 and 8.

1) because it is more complicated than what it looks like. What would be a fair comparison? Should we re-train all the models on the same dataset? should we use the same threshold value? what metrics should be used?

To answer your comment, No these values are not from text. We have run some of the models publicly available on our test set and have asked several authors to apply their models on our test set and report us their best results (as mentioned in the acknowledgement). We used the trained model (training dataset for each model is provided on tables 1, 2, and 3) first to test their generalization, affect of their training process, and the training size on the result. (This was mentioned in the comparison section). So the pre-trained model were used but on the same test set from STEAD. The manipulations were done by each authors and this why the tests were extremely time consuming.

(3) The discussion section wanders around and vaguely refers to supplement material leaving the reader to figure out exactly what is presented. Consider revising.

References to the figures in the supplementary materials are explicitly provided in the revision.

(4) The Japan analysis is good but add details. See below.

Major comments:

p10. "The shorter path through the detection decoder and its higher loss (due to longer length of label) naturally force the network to first learn to distinguish the earthquake signal within a time series." What does this refer to? The decoder layers are trained in parallel if my understanding is correct.

Yes, decoder layers are trained simultaneously but each has its own loss (a sigmoid as you can see from Figure 1). This is different than calculating the loss for three branches using a single softmax. The overall loss is a combination of weighted losses of these decoders. Because detection label is much wider it naturally results in a much higher loss compared to the P and S loss- that have much narrower picking labels. For instance for the picking if the network predict all of the points as zero still the loss would be much smaller than the detection because 5980 of 6000 samples in each picking label are zeros. Hence, because the biggest portion of the overall loss comes from detection, the network tries to minimize its loss first prior to the picking loss.

p11. Confusion matrix. 1 FP and 1-FN is impressive. What is used to determine a TP and TN? The softmax output will require a threshold to count as a TP; from supplement figure 4 the value is probably around 0.6. That said, every example shown is a detection softmax of ~1. Are there no situations where non earthquake signals appearing on all 3 channels produce a softmax prob of ~0.5, eg. the waveforms in Fig 4f occurs all the time in continuous records from non-tectonic signals. The noise used in model training is 200k samples and the earthquakes are 450k. How diverse is the noise? I suspect the model will show less confidence in the softmax predictions if tested with variable noise containing energy in the 1-15 Hz band, which is found in many environments.

There was no FN, the network produced only 1 FP. We used a threshold value of 0.5 for the detection based on precision-recall curves provided in Figure 4 of the supplementary materials. Also here we used Sigmoid not softmax. This has been depicted in Figure 1.

Yes there has been 1 example of noise with prob of > 0.5 which is the false positive in the confusion matrix. I have checked that sample and did not find any evidence of earthquake signal. The waveform in Fig 4f is an earthquake signal and the shown P and S arrival in top two channels are manually picked arrivals reported by USGS.

It is hard to quantify the diversity of noise samples. For us the most important factor during the completion of STEAD was to make sure the noise samples do not contain any earthquake signals. So we performed a designaling step. However, recently ~22000 samples of various anthropogenic noises with strong high-frequency energy were add to the STEAD which went through a manual quality control procedure. I agree with you that the model might exhibit lower confidence when it comes to the impulsive and high-energy noises. Check the new Figure 12 in the revision representing such noise.

Table 1. It is interesting that DetNet and Yews underperform to the classic averaging technique. In Yews (Zhu et al., 2019) they train with Wenchuan data only. Are the results here presented here using pretrained models from the authors and tested with STEAD? If so make that clear to show the other models do not generalize to seismic waveforms outside the study area. Or are the statistics taken from these studies.

This partially comes from the fact that the STA/LTA results reported here are based on a well-tuned STA/LTA. Unfortunately in the most of comparisons people usually only use the default parameters in the traditional algorithm resulting in poorer performance. The Georgia Tech group applied three Yews models on our test set one trained using Wenchuan dataset and two other models based on a larger training set in Taiwan. Here we report their best performance based on Taiwan model. Yes, their pre-trained model was applied on our test set from STEAD. We thought the training data and training size columns in tables convey this. However, following your suggestions I have added a sentence clearly mentioning this.

The discussion mentions the performance of Yews, GPD, and PkpNet is decreased because of windowing. Here, 1 minute windows are used. For continuous data how does the model handle events that occur near the edge of a 1-minute interval? How does the model handle a detection that is split by a window? How does the model handle small events occurring within seconds, i.e. multiple in 1 time window?

The non-normal distributions of errors for these models can be due several factors based on their training process which windowing is one of them. In our test set the arrival times of P-wave are clustered at the first quarter of the window while S-wave arrival are dispersed throughout the time window. All of the models resulted in a more or less normal distributions of errors for S-predictions. However, these three models had problems in predicting P-arrivals. A common point among these models is that all are using relatively wide labels for the labeling and prediction. For instance GPD does the perditions in 4 s windows and if the max of the perditions in a 400 sample window exceed a threshold it considers the middle of the window as the arrival time. Moreover, Yews considers that arrivals should be within a specific distance from beginning and end of a window. On the other hand, pickers like PhaseNet which uses much narrower window of 40 samples around the arrival times and perform extensive random shifting of the arrivals during the training does not show this problem. Anyway, it is hard to narrow down what exactly is the reason for this non normal distributions. Mentioning or the “moving window scheme” as a potential for this affect, was mainly based on our communication and consulting with the authors of these three models.

EqT handles the events occurring near the boundaries as long as there is a 0.2 s gap between the phase arrivals and the boundaries. This condition can always be satisfied when prediction is done in overlapping moving windows. Following are two examples of such cases occurred during applying EqT on continuous data in Japan:

I have added two new figures to the manuscript with multiple examples of such case. The model has been trained for local small-to-midsize events. So in most of the cases events fit in 1 min windows. However for longer waveforms the perdition can be done based on a portion of the waveform. This is one example:

The following figures are the results of applying our model to continuous data in Ridgecrest showing how the model handles cases with multiple events in a same window. This is due the augmentation and random addition of secondary events into each window during the training process.

A new figure (Figure 5) was added to the manuscript with multiple examples of such cases based on application of the model on continuous data from Ridgecrest and Tottori.

p15. It is difficult to assess why the p-waves are more uncertain than the s-waves. Supp. Fig 4 shows p and s waves with precision of 1 at all thresholds but a recall less than 1 for s wave threshold of 0.3 compared to 0.6 for p waves. A clearer presentation of how the detection threshold metrics are applied and reported is needed.

I do not understand why you interpret this as higher uncertainty in P-wave? In Supp Fig 4, we see that for the thresholds above 0.5, generally S-wave have lower recall compared to the P-waves which results in lower F-score as well. Lower recall is caused by higher false negative rates. Meaning that as we increase the threshold level we will miss more S-picks than P-picks. And this is simply because the network generally outputs relatively lower probabilities for the S-waves compared to the P wave. We can see this from Supp Fig 6 as well. And this can be interpreted as a higher uncertainty for S-picking. But why this happen? Because in general it is harder to pick S due to the overlying P-coda. This is true even for the human analysts and as the result we expect there is more picking error for S compare to P in our labels that are mainly based on manual picks. This explain why it is harder for the machine to learn S compared to P (we can see this from the learning curves in Supp Fig 7).

p16. It is not clear why the suggestion of using traditional pickers to improve the CNN picks is presented when Figure 7 shows EQTransformer having better statistics.

That was only a very general suggestion based on an approach that has been suggested in some works like Pardo et al (2019) suggesting to apply a secondary refinement step after initial picking by deep-learning. This means an aided version for traditional picker should perform better. Anyway I have removed it from the text in the revision.

The independent Japan analysis is good but details are lacking that would improve clarity. Per comments above: How does the model handle events occurring between window intervals? What threshold is set to determine a detection? Does cultural and environmental noise show increased softmax probabilities? How many false detections are removed by HypoInverse in the association of events? Is the increase in $M > 0$ events due to JMA staffing constraints that cannot spend time hand picking the smallest events? Overall the performance is good but 2-3 sentences of details would improve.

The code is able to handle these cases. It can be applied on overlapping windows and it automatically ignores the duplicate picks and only writes out the unique ones. I used threshold values of 0.5, 0.4, and 0.3 for detection, P-picking, and S-picking respectively. More than 85% of associated picks end up in the final catalog after the HypoInverse location and HypoDD relocation. However, it is hard to assess if the removed one were false detections or that was due to the error in the association. Note that we did not perform a sophisticated association algorithm here and simply considered the picks within a short time period as a single event. This can introduce errors by associating the picks of multiple events occurring in a short time period as a single event and vice versa. However, the super-high precision in the location and eliminating the fault structure is not the main goal. Here, the goal is to test the performance of a deep-learning model trained outside of a region and assess its limits. It is hard to say how and why JMA staff processed a dataset 20 years ago. However, thanks for the comment. I added a full paragraph explaining the details.

This is an example where both an earthquake and a high-frequency non-earthquake signal exist in a same window but the model did not output a high probability for non-earthquake pulse.

Minor comments:

All figure captions need more detail for the reader to understand the information presented.

I added more details to some of the figures and tables.

Figure 2 caption; check typo.

Fixed, thanks.

Figure 3. Very nice visual description for the attention layer but add axis labels and

convert x-axis to time (s), not index position.

What is the right axis? The summation of the weighted hidden states?

Done, no these are outputs of attention layers which are the summation of attention weights presented in supplementary materials.

Figure 4. Add detail to what is exactly show. Multiple channels I presume. Why is the predicted shown on one channel and manual picks shown on 2 channels? State what the confidence interval shown in the model prediction panel. It is not obvious it is there until 4f. Show the meta data for each events, e.g. magnitude and distance, to give the reader information of what is shown. Label each axis.

Done.

Christopher Johnson

Reviewer #2 (Remarks to the Author):

Review of the manuscript:

"Earthquake Transformer: An Attentive Deep-learning Model for Simultaneous Earthquake Detection and Phase Picking" by Dr Mousavi and colleagues,

The article provides a novel deep learning methodology to detect earthquakes and pick P- and S-wave onset phases. The technique is applied to the STEAD dataset and to the continuous data recorded in the aftershock region of the 2000 Mw 6.6 western Tottori earthquake. The results presented are remarkable and of great interest to a wide group of seismologists spanning from researchers in tectonics (larger number of located earthquakes resulting from lower magnitude detection thresholds) to operational seismic monitoring centers in charge of real-time earthquake detection and the compilation of national to global earthquake catalogs.

Overall, the adoption of the recurrent/transform/attentive scheme is novel and it appears to make a step forward towards an improved adoption of ML techniques in seismology. Because of the complexity of the methodology (several interconnected stages) and of the use of state-of-the-art solutions, the method section requires a solid background in the machine learning literature. Nonetheless, the addition of some detail on the choices made to design the methodology would be likely help the reader to follow and possibly replicate the proposed model and reproduce the results.

This is an important manuscript that should be published by Nature Communications. I recommend publication after minor revision.

Alberto Michelini (with the collaboration of Dario Jozinović)

Thank you very much for your comments and suggestions. I tried to address your all of your points during the revision.

General advice

It would be a good practice that throughout the article whenever there is made reference of the supplementary material it is also indicated what exactly therein (e.g., Fig. X of supplementary material, Supplementary Note Y, ...).

Done.

Main comments and questions

1. Additional details should be provided in the Training section. This should include the criteria used to select and optimize the network architecture and hyperparameters, i.e. how did the authors select the number of layers, the kernel size for the convolutional layers etc.

The network architecture design is based on domain expertise. Optimization and hyperparameter selection is done based on experiments on a large number of prototypes. We add this to the network architecture section.

2. It would be interesting to show some waveform examples in which the EQTransformer performs badly (which seems rare), to try to understand if there are any, and what are the limitations of EQTransformer.

I have added an example (Figure 12) where it triggered a false positive to the main text and discussed it in the discussion section.

3. Related to 2., it is not clear how the algorithm addresses cases in which there are more than 1 earthquake within the 60 s window. More specifically, since the training was done using the STEAD dataset which was constructed adopting only one (1) earthquake within the 60 s time window as criterion (cf 1st paragraph of the Earthquake Waveforms section of Mousavi et al., 2019). I have the following questions

This is correct that training data contains 1 event per trace. As was mentioned in the training section, during the training we performed multiple augmentation which one of them was to randomly add a secondary event to the empty part of the trace. This can be a scaled version of the same event or another event in the batch. This technique is very effective and as you can see from following figures resulted in application of the model to continuous data recorded in various region, it can handle multiple events easily.

A new figure (Figure 5) was added to the manuscript with multiple examples of such cases based on application of the model on continuous data from Ridgecrest and Tottori.

3.1. How were the time windows applied and shifted through the continuous data ? If overlapping windows were used, what was the shift in seconds ?

You do not need to shift the continuous data. All the augmentations mentioned in the training section are only applied to the training set (not even the evaluation or test set). This is to make the model invariant to statistical properties of the training set. For instance in the training set most of the P arrivals are occurring between 5 to 20 seconds from the beginning of the trace- this is to avoid inclusion of non-cataloged events into the training set. To make the model works on P arrivals at any time within the 1 minute window we randomly shift events within training traces. This is simply done by rotating the NumPy arrays with random lengths. As the results of this augmentation during the training the final model can pick events at any time as you can see from the provided examples. This is a one reason behind poor performance of some of the deep-learning models in picking P waves in our comparison.

3.2. What occurs when the window includes two or more earthquakes with the same 60 s window ? This is an important issue since the ability to detect and pick P and S phases in the early minutes of an aftershock sequence is crucial.

Please see my response to the previous comments.

3.3. At p. 10 there is provide a nice description of how the attention mechanism works and it would be important to see what occurs with the attention mechanism when two earthquakes are in the same window (i.e., the Fig. 3 with two earthquakes within the 60 s window). It would be important to add this special case amongst the different types of representative waveforms of Fig. 4.

I add examples with more than 1 events. However, visualization of the attention wieghts won't be informative for these events since these are very local events with very small s-p. Two new figures (Figure 5 and sub-figure 4) with several examples with multiple events have been added to the manuscript.

Other comments and questions

1 Introduction

P 2 bottom

“...0.01 second of error in determining P-wave arrivals can translate to hundreds of meters of error in location.”

This seems an exaggeration. Assuming an average $V_p = 6.0$ km/s, 0.01 s would result in no more than 60 m. Rather, there are other factors that can be more relevant including e.g. inclusion of exclusion of P and S phases which, when using not well calibrated local velocity models, can result into large changes of the earthquake focus.

Agree, revised the text.

P3 mid

The authors describe the basic constituents of the EQTransformer model and it can be helpful for the reader to have right away a reference to the basic literature that regards the attentive-based models (ie.g., ref. 47, Vaswani, et. al., 2017).

We have revised this paragraph based on suggestion of the third reviewer and added the mentioned reference.

2 Related work

P 5. Top

“... They trained and tested their network using earthquake and noise data recorded in Oklahoma (~2700 events and ~700,00 noise windows) and ...”

The test data set numbers used by Perol et al. (2018) are missing.

The following is what Perol et al. report in their paper

The test set contains all the windows for July 2014 (209 events and 131,072 windows of noise), whereas the training set contains the remaining windows (2709 events and

700,039 noise windows).

Thanks, revised the text.

3 Results

Data and Labeling

P 8 First paragraph:

Description is misspelled

Thanks, fixed it.

P 9 top

“...band-passed filtered between 1-45 Hz.”

What is the basis for choosing these values ? The STEAD data set contains also broadband data and the high pass corner (1.0 Hz) seems a little too high and it may be one of the reasons why the methodology performs more poorly at larger epicentral distances.

The filter band used in STEAD was to suppress the very low-frequency noise that were problematic for the several of our algorithms used for quality controls and making the new labels. However, we selected this band wide enough that do not limit the variety of potential applications that can be done using the dataset. STEAD is meant to be a general purpose dataset. Majority of the events have been recorded within 100 km and are high frequency. This is why the lower band of 1.0 Hz has been used. 45 Hz upper band was mainly to prevent the aliasing during the down sampling step.

P9 top

“...Triangular labeling ... adopted for the final model. In this form, probabilities of P and S are set to 1 right at the first arrival time and linearly decrease to 0 within 20 samples before and 20 samples after.”

Does this imply that effectively the same weight is given to the P phases regardless of the picking quality ?

Right, although STEAD include the pick weights, we did not find the reported weights from different monitoring agencies homogenous and reliable. So a same weight is assigned to all P and S labels. A more reliable scheme would be to estimate the aleatoric uncertainties of the picks through inference (similar to the approach used here <https://arxiv.org/abs/1912.01144>). However, this is extremely difficult for classification tasks and can be a separate study itself.

Training

See main comment #1

I added more description of augmentation process to this section.

Comparison With Other Methods

P11 first paragraph

“This can be seen clearly from the example in Figure 4-c, where two strong and apparently separate waves are detected as parts of a single event rather than two individual events.”

If the training was performed on a data set that featured only one earthquake per time window, is it possible that the model can detect two earthquakes in the same window ? Please clarify since this answers also the previous questions on multiple earthquakes in the same time window (main comment #3).

I provided detailed answer to this question in the previous comments. This is not due to the fact that training data only contain single event and heavily depends on the training process. Adding more examples with multiple events will hopefully clear this.

Tables 2, 3

Please verify the values for MAE (and MAPE).

According to the definition given in the supplementary material and assuming that the true values consists of arrival times reported in STEAD and the predicted are those resulting from the different models, the values of MAE appear too large. For example, if we consider e.g. 10000 sample errors with 0 zero mean and a standard deviation sigma of 0.3, the resulting value of MAE would be ~0.024.

In contrast, the MAE values for P and S picks of Figure 10 are perfectly consistent with the standard deviations provided.

Regarding the MAPE, since the true value ($t_{\text{catalogue}}$ in figs 6 and 7) appear at the denominator (see supplement), it is not clear how MAPE is determined.

Did the authors take the travel time as the true value ?

(The MAPE values are not provided for the Tottori earthquake sequence)

Thank you very much for pointing this. High MAE and MAPE values were due to the use of errors in terms of the number of sample difference between the true and predicted values instead of second. I recalculated these based on second to be more consistent with the reported mean and standard deviation and replaces both figures 7, 8 and tables 2 and 3.

P12

“A pick was considered as a true positive when its absolute distance from the ground truth is less than 0.5 second“

It would be useful to evidence this information also in the captions of Fig. 6 and Fig. 7.

Done.

P12 bottom

“...The error distributions for some of the deep-learning pickers are not uniform and

are not uniform and cluster at sporadic times due to their moving-window scheme. “
Can the authors elaborate a little more on this ?”

It is hard to find why this has happened exactly. The fact that this only happens for P and the error distributions for S look okay might be related to the difference in the distribution of the test data. The main difference is that P arrivals in our test set are mainly clustered between the first 5 to 20 seconds while the S waves are spreading through the entire time span randomly. However, this should not make a problem for the non-bias model that has learned to pick the arrivals at any part of the trace (as we see from the results of EqT, PhaseNet, and pickNet). What cause the other three models (GPD, Yews, and PpkNet) to have problems in picking the P arrivals is not fully clear to us. Based on communications with the authors of these three model we suspect that some hyperparameters in these models such as the size of moving window and its shift may play a role on this.

P14 end of the section

“Results are slightly better for S picks.”

This somewhat slight improved performance of the S wave picks observed for the Tottori earthquake sequence is intriguing. It is well known that S-waves onsets are much more difficult to pick and it is common to observe significantly larger std for S waves. This behavior is also observed with the STEAD data set and shown in Figures 6 and 7 for the EQTransformer and the other methodologies. Can the authors articulate their thinking on this observation and reconcile with the results obtained using STEAD ?

This comparison is done only on a portion of picks that were common between our results and those from JMA and indicates there is less divergence between our S picks and JMA’s. That can be due the fact that this subsample of the picks all are having a high SNR or characteristics that Japanese annalists did a very good job in picking them, On the other hand, there could be more emergent P arrivals with low SNR that make it difficult for either Our picker or JMA’s analyst to match the results.

Discussion

This section discusses the results using figures in the sduplementary material. For completeness of the main text, I would advise the authors to insert the figure(s) they consider the most significant into the main text.

I have revised the text and reference to the figures in the supplementary materials have been given explicitly now.

P14 1st paragraph

“A deeper network can result in more discriminatory power”

Correct but it is not known yet. It would be preferable to use “might” rather than “can”.

Here a general mapping power is meant. More layers, a more nonlinearity in the learnt mapping function. However, I applied your suggested change.

P14 2nd paragraph

“errors seem to correlate with noise level (supplementary information)”

There are several figures in the supplementary information regarding the errors and their variation according SNR, distance and magnitude. Which ones are the authors referring to ?

Figure 5, this was added to the text during the revision.

P14 bottom paragraph

“We did not find a clear correlation between the estimated epistemic uncertainties and picking errors.”

Can you articulate on what you consider the picking epistemic uncertainties

Epistemic or model uncertainties here are quantified by variations in the output probabilities. If you run the model multiple times on a same data you might get different output probabilities- this is the case mostly for noisier data that results in lower overall probabilities. Here we investigated a potential relationship with the amount of such a variations in the output probabilities and the errors between the predicted arrival time and the ground truth. And the results was that there is no obvious relation between these two.

P15 top

“...and the fact that P waves can be more difficult to pick when the first arrival is the diving wave Pn.”

This difficulty can also originate from the 1.0 Hz highpass filtering adopted. This should be mentioned.

I added this to the text, thanks.

P15 last paragraph

“...PickNet may have a similar problem since the training is done based...”

Figure 6 does not seem to show the problem described for the other ML automatic pickers. It may not be proper to mention it.

I removed those sentences.

P16 top

“Here we used estimates of theoretical arrival times to center the input windows around each phase as was suggested in...”

In this study, this approach was used only when training and not when testing. Correct ?

No their training was done based on manual picks, but their models only work on short windows centered around the phase arrivals. They use theoretical arrival times to roughly center these windows. So their models basically can only refine the theoretical

arrival time for events that their locations are known in the prior. I added this to the related work section.

P16 last paragraph

“These comparisons suggest there may be advantages to hybrid approaches in which deep-learning-based picks are refined either by applying a traditional picker or cross-correlation delay measurements to improve the results.”

It is not entirely clear what the authors want to say. Anyway, if the objective is to obtain cross—correlation picks, it may be just necessary to identify the P (and S) windows to be cross-correlated with a target waveform and it may not be needed the whole procedure described in this article which also seem to be very effective already.

That was only a very general suggestion based on an approach that have been suggested in some works like Pardo et al (2019) suggesting to apply a secondary refinement step after initial picking by deep-learning. Anyway I have removed it from the text in the revision based on your comment and the suggestions of other two reviewers.

P17 top

“...and a simple association method h. Using more advanced phase association algorithms (e.g. Glass3)...”

Would it be possible to have more information on the simple association method ? How may phases that were picked by EQTransformer were not associated to an earthquake ?

Or simple association was mainly grouping the picks in a very short time windows (25 s). This rough and naïve association is prone to false association of phases of multiple events to a single event which results in rejection of the event by location code. Here we just suggest that ~ 10-15 % events that were rejected during the location and relocation process may be actual events that our simplistic associator was not able to correctly identified and associate their phases.

Methods

Very deep encoder

“We build a very deep encoder, which is known to be important in performance of a sequence-to-sequence model with attention.”

Related to main comment #1 and more specific, have the authors tried with a not as deep encoder ? It would be important to know if within the same scheme a deep encoder is really needed.

This goes back to the optimization procedure. Yes we have tried both shorter and deeper networks and came up with the current depth for the size of our dataset based on both the training loss and training time. In the provided code you can simply specify depth of the decoder and train and test different models.

Attention Mechanism

P19

The definitions of n and d_h are missing.

Definitions were added.

P20

"We replace the multi-head scaled-dot product attention by above single-head additive attention Figure 14."

Can you explain why you replace it ?

Multi-head are multiple attention layers in parallel. They are really time consuming to train. Their usage in the original Transformer paper was to add more mapping power since the Transformer does not have any typical convolutional layer. This is why I modified it to speed up the training and also make it easier to visualize the attention weights.

Supplementary material

Note 2. I did not find the Catalog of all events detected in the material downloaded

I add the catalog to the supplementary materials.

Supplementary movies. Probably just one movie is enough. The focus of the article is not the Tottori earthquake sequence but the new EQTransformer technique.

Only three movies were kept in the revision.

Figure 2 (or 3). Please add case with two or more earthquakes within the same 60 s window.

Two new figures (Figure 5 and sub-figure 4) with several examples with multiple events have been added to the manuscript.

Figure 5. Are there used two different color scales for P and S absolute errors ? Is the color proportional to the number of counts for each pixel ?

Yes, values for P are shown in blue and for S in red in the background while their density (or counts) are depicted with color coded contours. I added this description into the figure captions.

Reviewer #3 (Remarks to the Author):

This manuscript documents a novel deep-learning approach for local/regional earthquake detection and P and S wave picking. The approach uses an attentive model which allows it to focus on local features (P and S wave arrivals) to improve upon arrival-time picking following detection. The model is unique in that it is trained using data from a set of global earthquakes. The manuscript does an extensive job of comparing the performance of the model as compared to both traditional pickers (e.g., sta/ta) and previously published deep-learning approaches. The approach is further validated by testing its performance on detecting and picking a sequence of earthquakes in Japan. I've attached a marked-up document with the majority of my comments.

The authors do a good job of describing the model and it is clear that this represents the cutting edge of the application of deep-learning to earthquake detection and picking. The comparison of this model with other methods is above and beyond what is typically done and emphasizes the improved performance of the model architecture/training employed. Overall, I have few comments in regard to the work presented here, although I would have liked to see more description on how it is applied to continuous data.

I have added two full paragraphs (in pages 9 & 10) explaining how the model can be applied to continuous data. Moreover, provided link (in the foot print of page 3) contains a package of codes for applying the model to continuous data. It contains auxiliary modules that can automatically download continuous data for a time period and specific region, pre-processed continuous data in the conventional formats, apply the model and generate outputs. The package contains modules for building a model and testing it as well. All the results in this study are reproduceable using provided codes. A full tutorial and pre-trained model will be added to the GitHub page as well.

I think the presentation, organization, and resulting impact of the manuscript could be improved. The manuscript is written expecting a fairly high level of understanding of machine learning, and to most seismologists, I expect it will not be clear how this is a significant step forward in the application of these technologies to earthquake monitoring.

I have tried apply some changes to the organization of the paper and more explanations.

In terms of organization, there is a large portion of the introduction that details previous works, highlighting the distinct architectures of each approach. I think this level of detail will go beyond most readers and distracts from highlighting what makes this approach unique. Some of the features highlighted in the model (in the six main points) will also go beyond most readers and in themselves are not as important as the improved performance. (e.g., why does it matter that this is the deepest model? why does it matter that this is the first attentive model? why does it matter that it combines recursive and CNN architectures?).

One of our goal in this study was to provides a short summary of what have been leaning from all of these efforts in making deep-learning detection/phase picking and highlight the elements that might play a more important roles. This is why here we provided more details of previous studies than is usually given and dedicated a whole section to it. Providing such section and this level of details is a norm and a good habit in the ML community that seismological community might benefit from adapting to it. This can help readers in the following section where comparison with other models are performed and discussed. However, based on your suggestion I have moved the “related work” section to after the discussion section toward the end of the manuscript. I have moved the six points to the conclusion section as well.

The results section discusses the architecture and training of the model, but I think this should be placed entirely in the methods section. Instead, I think the results should immediately highlight the improvements of the model as compared to previous methods and other important features like its huge leap in computational efficiency. The application to continuous data is important and I don't think it is highlighted as it should be. If the authors could point to what new observations come from creating such a high-quality dataset and how efficiently they are able to produce it, I think it would strengthen the impact of the paper significantly.

Since part of the results and exploring the attention weights requires a knowledge of the network architecture we couldn't move all of these sections to the methodology at the end of the manuscript. On the other hand moving the methodology section with all of the details to the front does not make sense. Hence, I divided the result section in the previous version into two sections of “Model Building” and “Results” in the revised MS.

The reported computational times are the running time of the whole prediction algorithm on pre-processed data (filtered continuous data sliced into 1-minute windows). This includes the time for writing out the detection/picking results, generating a number of plots, writing out all the perdition probability. This highly depends on the number of existed events in the continuous data. The reported computational time for one month of continuous data in the text is based on the application to the Japanese dataset and its level of seismicity. For a same duration of continuous data with much less earthquakes this time can be much lower (< 10 min). Moreover, this is without uncertainty estimation that requires multiple perditions and can significantly increase the computational time. The lower computational time of deep-learning models compared to other traditional methods is well known and have been pointed out by multiple studies before. We did not emphasize on this part since more analyses are need to accurately discussed the leap in computational efficiency. It is really hard to convey such tests at current time due to the complexity of or algorithms. Time complexity can significantly change based on implementations. However, we will dig more deeply into this aspect in our next study in near future.

I am a little concerned over the comparisons to other methods, but I am thrilled to see the attempt. It's a difficult problem because each deep-learning model was trained with

distinct applications in mind and because more general methods can be optimized for specific use cases. I think the authors should expand upon the criteria used for the different standard pickers (filtering, STA/LTA windows, etc.). The testing set used to validate the models is a subset of the overall data set. Therefore, it was selected following the same criteria as the training data. This means that there could inherently be some bias that would improve the apparent performance of their model.

The distribution of the training set has been significantly disturbed during the training process by heavy augmentation. And the application on the Japanese dataset shows that the good performance of the model is not limited to the test set that comes from the same dataset. On the other hand, a very good performance of models like PhaseNet that were trained on different dataset suggests that the training set does not play as important role as the training process plays. (PhaseNet also takes advantage of some augmentations). Overall, I agree with you that a test on totally independent set would be more fair. However, it is hard to find a bench mark set with a high number of reliably labeled samples that enables a statistically meaningful tests. I have consulted with the authors of (GPD, Yews, and PpkNet) on the weird error distributions. Moving window effect was based on their thoughts. However, they all agreed that it is hard to narrow down the main reason why this happens. I have added a whole paragraph (before Figure 13) explaining all of these challenges in the comparison.

Some quick points/questions not in the PDF:

What happens when you have signals from events beyond 300 km (e.g., teleseisms?)

Did not tested on such cases, so cannot answer. I should add although we have samples up to 300 km in our training set but majority of them are within 100 km.

From supplemental figure 5, it looks like your data is dominated by events within 100 km? Why are there so few records beyond 100 km?

That is correct. Majority of the samples in STEAD are events with magnitudes up to 2.5 M and epicentral distances of up to 100 km. This is due to the conditions we considered during building of the dataset. The use of 1 min window was to initially avoid inclusion of un-cataloged events. On the other hand we set up some criteria to only keep events that most of their full-waveform (from 5 s before P to 10 s after S) fits into this 1 minute window. This excludes most of the data with either large magnitudes or epicentral distance due to their long waveforms. This would not affect our initial focus which was concentrated on small and local seismicity which is where the uncomeliness of most of our catalogs come from. This will hopefully be improved in the next version of STEAD which will include longer windows of larger events (>2.5) within 100 to 600 km range. I have added some more information on the dataset property into the "data and labeling" sub section.

Somewhere in the figure/table captions, you should mention MAE is in samples as opposed to seconds.

Thanks for pointing this. I have recalculated MAE and MAPE based on seconds and replaced figures and values in the tables.

I don't think the movies are necessary at the moment because the tectonic interpretation is not highlighted in the paper.

Agree, I cut them into 2-3..

Long term, have you considered trying to isolate Pg vs Pn, Sg vs Sn at these regional distances?

No, as I mentioned in the previous comments our current dataset is not rich in distances over the turn over range and we kept only the direct phased during the labeling of STEAD. It could be a useful and interesting project in the long term but is not in my to do list at the moment.

I have addressed almost all of the comments on the annotated PDF. These are the answers to the remaining questions on the PDF:

STEAD does not contain Japanese data currently mainly because of some concerns regarding the redistribution policies that are different in JMA compared to IRIS or other International agencies.

45 Hz seems much higher than necessary. I take it your intent is to filter up to the nyquist Hz more or less. Why not just apply a high-pass? How much have you explored filtering?

The filter band used in STEAD was to suppress the very low-frequency noise that were problematic for the several of the algorithms used for quality controls and making the new labels. However, we selected this band wide enough that do not limit the variety of potential applications that can be one using the dataset. STEAD is meant to be a general purpose dataset. 45 Hz upper band was mainly to prevent the aliasing during the down sampling step.

This seems overly restrictive. Are S picks at 3 degrees by a human this accurate? I know Sn picks can have a spread of residuals on the order of 5 s (but that includes earth model issues). Still, I expect that humans can't pick Sn within 0.5 s, so their is inherent uncertainty in the validation dataset.

I totally agree that it is hard to define a set of criteria and matrices to measure the performance. In the previous studies (e.g. Pardo et al. 2019 or Zhu and Beroza 2019)

even a shorter window of 0.2 s were used. I think 0.5 should be a good range for direct waves especially when they are recorded in epicentral distance of less than 100 km.

I know in GPD for example, the sub sampling is settable parameter (i think), so you could fix this issue by having it run on every sample.

The non-normal distributions of errors for these models can be due several factors based on their training process which windowing is one of them. In our test set the arrival times of P-wave are clustered at the first quarter of the window while S-wave arrival are dispersed throughout the time window. All of the models resulted in a more or less normal distributions of errors for S-predictions. However, these three models had problems in predicting P-arrivals which a common point among them is that all are using relatively wide labels for the prediction. For instance GPD does the perditions in 4 s windows and if the max of the perditions in a 400 sample window exceed a threshold it considers the middle of the window as the arrival time. Moreover, Yews considers that arrivals should be within a specific duration from beginning of the window during its training. On the other hand, pickers like PhaseNet which uses much narrower window of 40 samples around the arrival times and perform extensive random shifting of the arrivals during the training does not show this problem. However, it is hard to point out exactly the reason behind this. I have consulted with the author of GPD (Zachary Ross) on this.

You think a traditional picker should be applied to refine deep-learning picks? Isn't your picker out performing the traditional methods?

That wan only a very general suggestion based on an approach that have been suggested in some works like Pardo et al (2019) suggesting to apply a secondary refinement step after initial picking by deep-learning. The idea is that an informed unsupervised method like traditional pickers would significantly perform better. Anyway I have removed it from the text in the revision based on your comment and the suggestions of other two reviewers.

Can you expand on how you came to this model. Did you do a large search through a wide range of architectures?

Right, it is based on a large search and experimenting with many network with various hyper parameters. I add a sentence into the "network architecture" section on this,

REVIEWERS' COMMENTS:

Reviewer #1 (Remarks to the Author):

The authors responded to all questions/comments raised and did a good job with all reviews. One reviewer's suggestion to restructure the manuscript resulted in sections being moved around. Please consider double checking Supp fig order and section headers.

Minor comments are provided in the pdf.

I recommend publication and to address these minor changes in the text.

Christopher Johnson

Reviewer #2 (Remarks to the Author):

The revised manuscript is fine and it can be accepted as is.

Reviewer #3 (Remarks to the Author):

The authors have done a good job at addressing my previous comments, and therefore I am happy to recommend the manuscript for publication. Particularly, the changes to the organization of the manuscript and the added descriptions have improved the manuscript's readability, accessibility to a broader audience, and I think will improve the impact of paper. Below are a few small suggestions.

Page 9: "The model works very well for earthquakes with different magnitudes, epicentral range, and waveform shapes." - I am not sure you can say this so generally. Your test dataset is near-source and relatively small magnitude.

Page 10, 2nd Paragraph: I think it would be good to add a sentence here acknowledging that there is a level of tuning involved in each of these approaches, and that the performance can vary based on this tuning. This is true for deep learning pickers too, such as assigning a softmax threshold for pick declaration.

Page 13: "former was able to pick 401,566 P and S arrival time on 18 of those stations (due to unavailability of data for other stations)" - I think it is worth pointing this out earlier in this section. You are creating a catalog from 2/3 less stations than what was used in the comparison catalog. This makes the improved performance even more impressive. - You do mention this on Page 17.

Page 14, 2nd Paragraph: The false positive rate is low compared to standard methods (STA/LTA). Therefore, the chances of falsely associating picks from multiple stations into an event is low. On the other hand, in the false positives you show in figure 14 both a false P and S are predicted. I imagine that necessitated by the design of the model, and there will always be a false P and S pick when a false detection is made, resulting in two false picks.

Will Yeck

REVIEWER COMMENTS

Reviewer #1 (Remarks to the Author):

The authors responded to all questions/comments raised and did a good job will all reviews. One reviewers suggestion to restructure the manuscript resulted in sections being moved around. Please consider double checking Supp fig order and section headers.

Minor comments are provided in the pdf.

I recommend publication and to address these minor changes in the text.

Thank you for your comments and suggestions. I addressed all the comments on the PDF.

The reference to the supplementary figure on page 8 has been changed.

The title of subsection "comparison with other methods" as well as a few sentences in the subsection have been revised.

Reviewer #3 (Remarks to the Author):

The authors have done a good job at addressing my previous comments, and therefore I am happy to recommend the manuscript for publication. Particularly, the changes to the organization of the manuscript and the added descriptions have improved the manuscripts readability, accessibility to a broader audience, and I think will improve the impact of paper. Below are a few small suggestions.

Thank you for your comments and suggestions. I tried to address your comments in the revision.

Page 9: "The model works very well for earthquakes with different magnitudes, epicentral range, and waveform shapes." - I am not sure you can say this so generally. Your test dataset is near-source and relatively small magnitude.

Right, of course we meant the magnitudes and distances within the test set ranges. However, to address your comment and avoid future misinterpretations, I have changed the sentence to: "the network predictions for 6 representative samples from the test set." which implies the size and distance ranges implicitly.

Page 10, 2nd Paragraph: I think it would be good to add a sentence here acknowledging that there is a level of tuning involved in each of these approaches, and that the performance can vary based on this tuning. This is true for deep learning pickers too, such as assigning a softmax threshold for pick declaration.

One sentence was added to acknowledge this.

Page 13: “former was able to pick 401,566 P and S arrival time on 18 of those stations (due to unavailability of data for other stations)” – I think it is worth pointing this out earlier in this section. You are creating a catalog from 2/3 less stations than what was used in the comparison catalog. This makes the improved performance even more impressive. – You do mention this on Page 17.

One sentence (These are a portion of stations (57) originally used for studying this sequence by the Japan Meteorological Agency (JMA).) was added to the beginning of the section. I also emphasized it a bit more on page 17.

Page 14, 2nd Paragraph: The false positive rate is low compared to standard methods (STA/LTA). Therefore, the chances of falsely associating picks from multiple stations into an event is low.

One the other hand, in the false positives you show in figure 14 both a false P and S are predicted. I imagine that necessitated by the design of the model, and there will always be a false P and S pick when a false detection is made, resulting in two false picks.

Not necessarily. Although we tried to bound the detection and picking task on each others through the attention mechanism, their learnings are still based on separate loss functions and to some extent they preform independently. So it is possible to have some false positive detections but without any picks depending on the choice of threshold. Anyway it would be extremely rare to have a false detection with false picks with high probabilities and low uncertainties.